# Artificial cellulosic leaf with adjustable enzymatic CO$_2$ sequestration capability

Xing Zhu [1,2,5] ✉, Chenxi Du[1,2,5], Bo Gao[3] & Bin He [1,4] ✉

Developing artificial leaves to address the environmental burden of CO$_2$ is pivotal for advancing our Net Zero Future. In this study, we introduce EcoLeaf, an artificial leaf that closely mimics the characteristics of natural leaves. It harnesses visible light as its sole energy source and orchestrates the controlled expansion and contraction of stomata and the exchange of petiole materials to govern the rate of CO$_2$ sequestration from the atmosphere. Furthermore, EcoLeaf has a cellulose composition and mechanical strength similar to those of natural leaves, allowing it to seamlessly integrate into the ecosystem during use and participate in natural degradation and nutrient cycling processes at the end of its life. We propose that the carbon sequestration pathway within EcoLeaf is adaptable and can serve as a versatile biomimetic platform for diverse biogenic carbon sequestration pathways in the future.

Carbon dioxide (CO$_2$) fixation through tree leaf photosynthesis is the most critical natural process on Earth, playing a pivotal role in maintaining ecological balance[1,2]. However, in the context of accelerating global industrialization, CO$_2$ emissions are escalating, overburdening photosynthesis and posing a significant threat to ecological equilibrium. In light of this scenario, the pursuit of artificial foliage for efficient carbon sequestration has become a pressing concern[3–5].

The first generation of artificial leaves primarily focused on catalyzing hydrogen production from water decomposition using metal catalysts such as platinum[6], ruthenium[7,8], and nickel[9] in the presence of light. This reaction, akin to photosynthesis in natural leaves, transforms light energy into a form that humans can utilize. However, this first-generation technology did not contribute to carbon sequestration or address the ecological imbalance resulting from CO$_2$ emissions. By contrast, the second generation of artificial leaves involves the development of photoactive integrated components, configured in wired and wireless setups, by depositing photochemical and chemical catalysts onto semiconductor substrates like InP[10,11], ZnTe[12–14], Cu$_2$O[15,16], p-Si[17,18], and encapsulated photovoltaic perovskites[19–21], primarily through deposition techniques. These components primarily utilize sunlight as their primary energy source, orchestrating a series of redox reactions to collectively convert CO$_2$ into organic compounds (e.g., organic acids, aldehydes, alcohols, olefins, polysaccharides),

emulating the process of artificial natural photosynthesis. Nevertheless, existing photoelectrochemical (PEC) systems exhibit limited light energy utilization, intricate preparation procedures, and reliance on costly raw materials. Furthermore, the catalysts employed in most PEC systems, encompassing photochemical, electrochemical, and chemical catalysts, are less selective, susceptible to side reactions, and significantly less catalytically efficient compared to biocatalysts.

The crux of natural leaf photosynthesis lies in the catalytic effect of enzymes[22,23]. In contrast to other catalysts, enzymes offer distinct advantages, including high efficiency, specificity, mild reaction conditions, and a renewable source in the form of green biological resources[24]. With the advancement of biotechnology, large-scale enzyme production at a low cost is attainable through microbial fermentation, making CO$_2$ fixation through biocatalysis the focal point of the third generation of artificial leaves. Erb et al. devised a microfluidic system emulating plant chloroplasts, leveraging cyst-like membranes from spinach to replicate the photosynthetic process[25]. This system enables a cyclical process involving crotonyl-coenzyme A (CoA), ethylmalonyl-CoA, and hydroxybutyrate acyl-CoA (CETCH), facilitating CO$_2$ fixation and subsequent photoconsolidation reactions within cellular-sized water-in-oil droplets. This breakthrough marks a significant stride in carbon cycling as it harnesses enzymes as catalysts to capture and convert CO$_2$. Cai et al. employed a computational pathway

[1]College of Bioresources Chemical and Materials Engineering, Shaanxi University of Science & Technology, Xi'an 710021, China. [2]Institute of Biomass & Functional Materials, Shaanxi University of Science & Technology, Xi'an 710021, China. [3]School of Chemical Engineering, Northwest University, Xi'an 710127, China. [4]Key Laboratory of Paper Based Functional Materials, Shaanxi University of Science & Technology, Xi'an 710021, China. [5]These authors contributed equally: Xing Zhu, Chenxi Du. ✉e-mail: zhuxing@sust.edu.cn; prof.hebin@sust.edu.cn

design to achieve a thermodynamically favorable reaction pathway for artificial starch synthesis[26]. By assembling and replacing 11 modules consisting of 62 enzymes from 31 organisms, they established the enzyme-catalyzed metabolic pathway for artificial starch synthesis (ASAP) 1.0. This research successfully increased the rate of starch synthesis through ASAP, exceeding maize's capacity by a factor of 8.5, thus opening new possibilities for $CO_2$ fixation and reuse. However, these studies primarily focused on artificial $CO_2$ conversion pathways, with the systems' structures and functions diverging significantly from those of natural leaves. For example, natural leaves comprise lignocellulose, a material that is natural, environmentally friendly, easily degradable, and capable of nourishing the soil upon falling[27]. They also possess a stomatal structure, which enables the regulation of carbon sequestration rates[28]. Additionally, natural leaves feature conduits for water and metabolite transport, ensuring that sequestration products do not accumulate and that sequestration reactions continue uninterrupted[29]. These functionalities play a crucial role in realizing the carbon cycle process. Hence, to achieve ecological rebalancing, developing artificial carbon sequestration pathways and corresponding biomimetic functionalization platforms represents an intriguing yet formidable challenge.

In this work, inspired by the ancient Chinese myth of Ma Liang, a painter capable of bringing his drawings to life with his magic brush, we paints leaves on paper and endow them with the essential functions of natural leaves, including light trapping, carbon sequestration, stomatal regulation, and material transport. This endeavor encompasses several key innovations: (1) A three-dimensional mesh matrix with photoisomerization properties is meticulously constructed on the fiber substrate through controlled radical graft polymerization. Within this matrix, carbonic anhydrase (CA), an enzyme catalyst pivotal for $CO_2$ sequestration, is embedded. When exposed to specific wavelengths of light, the azo bonds in the molecular structure of the 3D mesh matrix undergo cis-trans isomerization, thereby endowing the Ecoleaf with adjustable stomatal size characteristics essential for carbon sequestration similar to natural leaves. When the stomata are expand, $CO_2$ can efficiently interact with CA, being swiftly captured within the Ecoleaf. When the stomata contract, the transmission path between external substances and CA is obstructed, and the denser mesh size exerts a binding effect on the CA enzyme's protein structure, enhancing stability and environmental resilience. In addition, similar to the dark reaction in natural leaves, stomatal constriction facilitated a more stable catalytic conversion of carbon sequestration products by subsequent cascade enzymes in EcoLeaf. (2) By creating, cutting, spraying, and dyeing EcoLeaf with varying K/S values (color depth), we replicate the ability of natural leaves to harness dark leaf epidermis for rapid and efficient light energy capture, subsequently converting it into heat energy, which powers CA, the primary carbon sequestration organ. (3) Exploiting the principle of osmotic differences, connecting the petiole of EcoLeaf to water facilitates the diffusion and recycling of carbon products and water. This process ensures a continuous proton source required by CA, a cornerstone for gaseous $CO_2$ capture, while preventing the inhibition of CA's carbon capture ability due to the accumulation of leaf products, thereby rendering carbon capture behavior sustainable. (4) EcoLeaf, which uses natural cellulose paper as its main raw material, achieves strong mechanical and biodegradation properties. It represents a robust, and sustainable artificial leaf that is amenable to large-scale production and application. Our work aims to provide a biomimetic functionalized platform for artificial carbon sequestration pathways that is versatile in accommodating various biosequestration reactions, offering the flexibility to meet diverse usage requirements.

## Results and discussion

In this study, we constructed a biomimetic functionalized leaf, as depicted in Fig. 1a, b, beginning with a leaf drawn on paper. This artificial leaf, similar to natural leaves, integrates essential properties for the natural photosynthetic and carbon sequestration processes, including light capture, stomatal expansion and contraction, gaseous $CO_2$ capture, product transportation, and biodegradability.

A natural leaf primarily consists of the leaf blade plane, leaf flesh, and leaf veins[30,31]. The leaf plane is typically flattened to maximize sunlight reception[32]. Leaf pulp forms a three-dimensional reticular matrix responsible for organic matter production and storage through photosynthesis[33]. Leaf veins facilitate substance absorption, transportation, and distribution[34]. Inspired by this, in our study, we cut paper-drawn leaves into EcoLeaf planes. We covalently attached glycidyl methacrylate(GMA) with a C=C bond to both ends of the photoresponsive monomer 4,4′-Azodianiline(4,4′-AZO) through covalent conjugation. This GMA/4,4′-AZO was grafted onto the surface of the EcoLeaf planes, creating a photoresponsive three-dimensional reticulated matrix under visible light. The synthesis of the photoresponsive monomer GMA/4,4′-AZO was validated using NMR to confirm the epoxy and amino group reaction. The disappearance of the characteristic 4,4′-AZO amino proton peak (5.80 ppm) and the emergence of the nascent hydroxyl proton peak (5.2 ppm) of GMA/4,4′-AZO (Supplementary Fig. 1b–d) indicated the successful epoxy ring-opening addition reaction. This provided theoretical support for the attachment of the 3D mesh matrix to the cellulose substrate surface. A comparison between Fig. 1c, d demonstrates that graft polymerization of GMA/4,4′-AZO on the surface of kapok fibers fills the pores between fibers with a light-responsive reticulation matrix. This results in a denser fiber network, initial evidence of the successful construction of a three-dimensional reticulation matrix on the cellulose substrate.

Natural leaves rely on mitochondrial respiration for carbon sequestration, a process facilitated by enzyme complexes with high catalytic activity and selectivity. In our study, CA, known for its efficient $CO_2$ turnover, served as the primary carbon sequestration organ in EcoLeaf. To confirm CA's integration into the artificial leaf system, we encapsulated free CA with FITC staining and observed the distribution of FITC-CA within the artificial leaf system using LSCM. The green fluorescence in Fig. 1e is attributed to the chromophore group (-N=N-) in 4,4′-AZO, affirming the successful polymerization of the photoresponsive monomer on the cellulose surface. Additionally, the presence of numerous bright green fluorescent clusters in Fig. 1f, compared to Fig. 1e, can be ascribed to FITC-CA. To further visualize the 3D mesh matrix and CA, we explored the 3D distribution state of both in EcoLeaf using LSCM. Supplementary Fig. 2 displays the 3D tomograms of EcoLeaf and the 3D maps from different top-down view angles (0°, 60°, 90°). The encapsulation of the fiber surface by the 3D mesh matrix and the filling of the fiber pores can be clearly seen in the figure, reinforcing the successful construction of the 3D mesh cloth matrix. Additionally, the bright green fluorescence (FITC-CA) exhibits a relatively uniform distribution within the 3D mesh matrix. FITC contains a benzene ring and a thiocyanine group, which shift fluorescence emission peaks to longer wavelengths, thus intensifying fluorescence. By contrast, the relatively simple molecular structure of azobenzene in GMA/4,4′-AZO leads to weaker fluorescence. This preliminary evidence confirms the successful inclusion of CA in the EcoLeaf system. Furthermore, XPS analysis was conducted at various stages of EcoLeaf preparation to examine elemental and covalent bonding changes on the material's surface. In comparison to Cp (Fig. 1g), the C $1s$ core spectrum of Cp-ITX and the full spectrum (Fig. 1h) show new C-S bonds and S introduced by the photoinitiator ITX. Additionally, the newly formed C-N bonds in Fig. 1i primarily originate from GMA/4,4′-AZO and CA. O-C=O and N-C=O are associated with the peptide bond and carboxyl group of CA, providing further evidence of successful EcoLeaf construction.

The mechanical properties of natural vane play a crucial role in its ability to withstand external forces. Good mechanical properties allow the blades to maintain their structural integrity and prevent them from

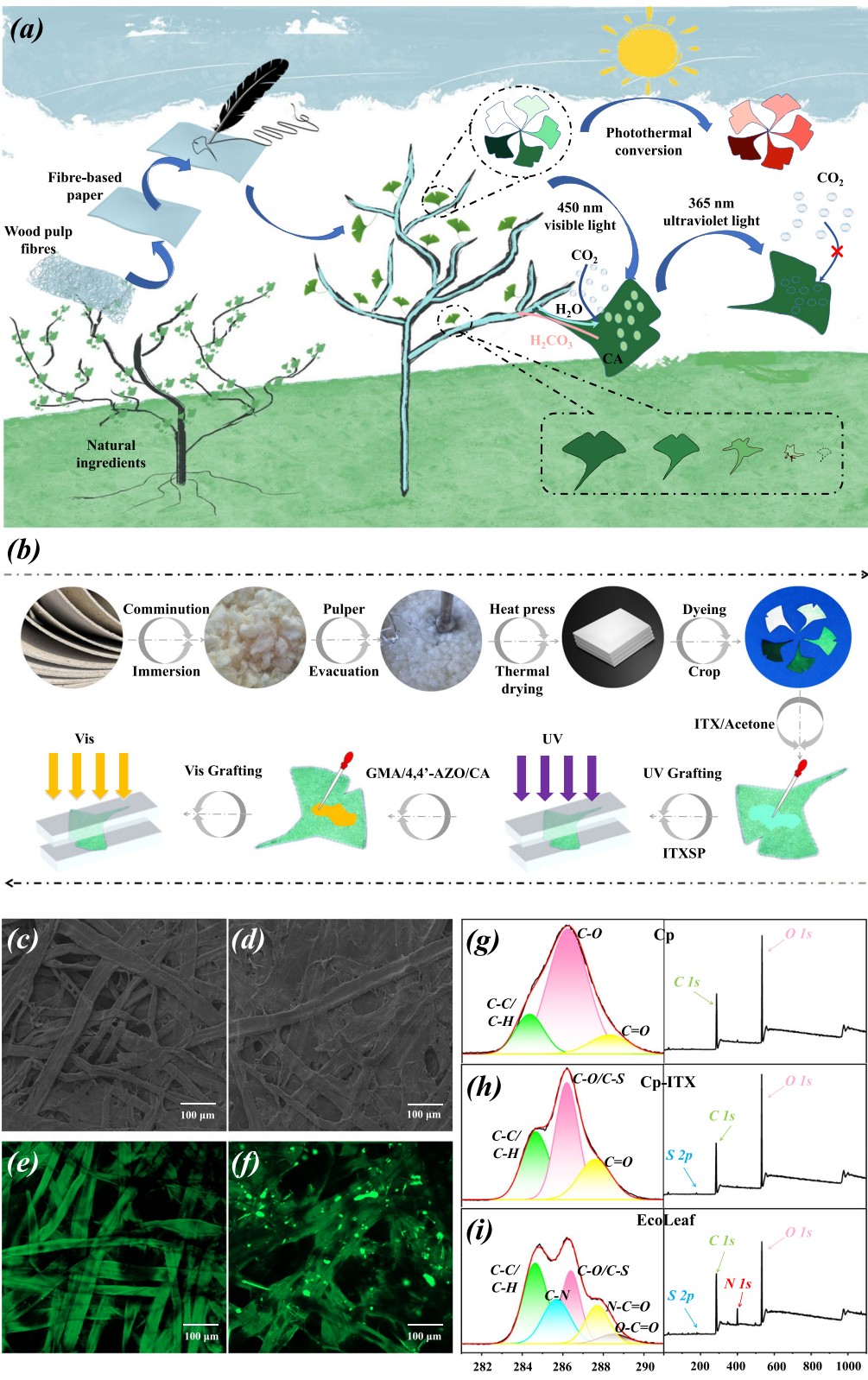

**Fig. 1 | Preparation of EcoLeaf and its structural characterization.**
**a** Construction and performance of EcoLeaf; **b** The detailed preparation process of EcoLeaf; SEM maps of light-responsive 3D mesh matrix (Isopropyl thioxanthone, ITX) (**c**) before grafting vs. (**d**) after grafting of cellulose substrate; Fluorescence spectra of (**e**) EcoLeaf without encapsulated CA vs. (**f**) EcoLeaf encapsulated with FITC-CA under LCSM (The experiment was repeated independently three times with similar results); C 1*s* core-level spectra and full spectra of (**g**) Cellulose paper (Cp), (**h**) Cellulose paper-isopropyl thioxanthone (Cp-ITX), and (**i**) EcoLeaf.

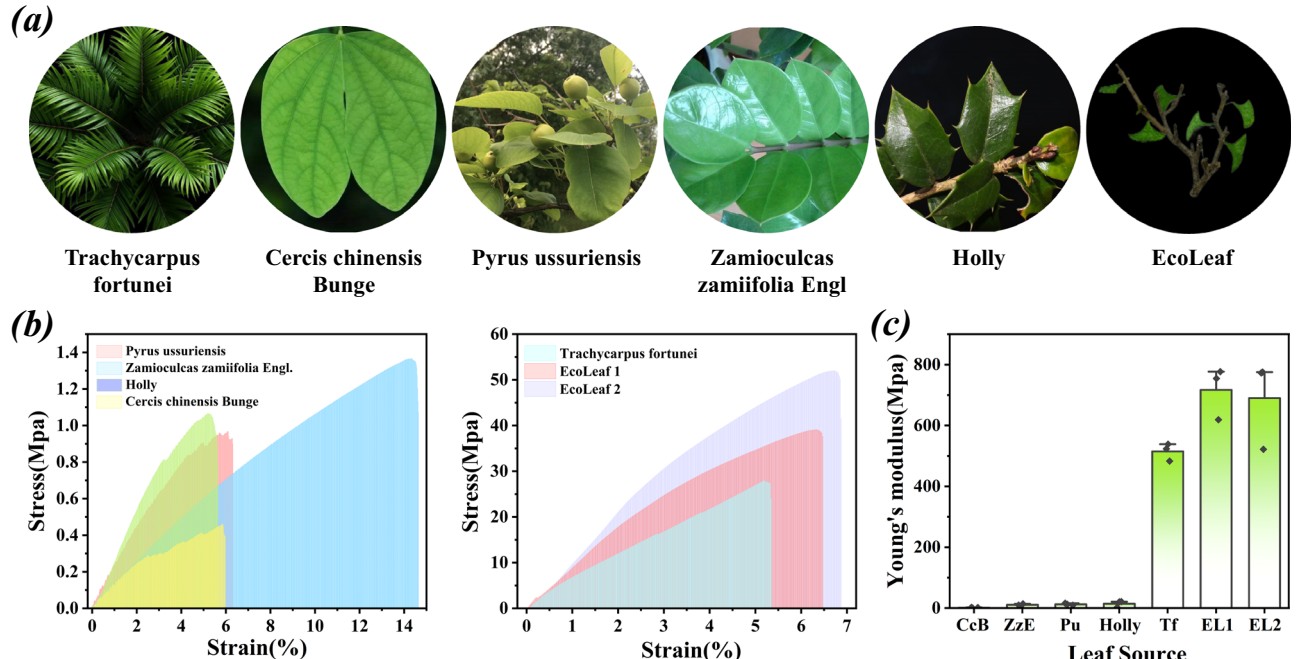

**Fig. 2 | Comparison of mechanical properties of natural blades vs. EcoLeaf. a** Natural leaves from *Trachycarpus fortune* (Tf), *Cercis chinensis* bunge (CcB), *Pyrus ussuriensis* (Pu), *Zamioculcas zamiifolia* engl (ZzE), *Holly*, and EcoLeaf (EL) respectively; **b** Stress-strain curves and **c** Young's modulus of natural leaves from different sources and EcoLeaf (EL 1 thickness 0.19 mm and density 0.60 g/cm³, EL 2 thickness 0.22 mm and density 0.61 g/cm³) (Parallel experiments with three sets of identical samples, data are presented as mean values +/−SEM).

breaking, getting damaged, or deforming under the influence of wind, gravity, rain, snow, and frost. Based on this, the mechanical properties of five different natural leaves and two EcoLeafs with different densities and thicknesses were investigated (Fig. 2a). It is evident that the mechanical properties of the paper used to make EcoLeaf are higher than those of most natural leaves in the left panel of Fig. 2b. Furthermore, based on the findings in Fig. 2b, c, it can be observed that the palm fronds exhibit slightly lower ultimate stress and Young's modulus values compared to EcoLeaf, despite their inherent strong mechanical properties, which can be primarily attributed to the cross-linking effect of its 3D mesh matrix. The manufacturing process of EcoLeaf leads to a more homogeneous and optimized structure, resulting in higher mechanical properties. This means that EcoLeaf is more resistant to deformation and can withstand greater forces without breaking or shattering, making it more environmentally tolerant. Mechanical advantages make EcoLeaf a promising alternative to natural leaves for various applications requiring strength and durability.

The efficiency of light reaction, a pivotal process in plant photosynthesis, is influenced by factors like chlorophyll species and content, chloroplast structure, and the catalytic activity of photosynthetic enzymes. Among these, photosynthetic enzyme activity plays a crucial role in a plant's overall photosynthetic rate. This enzyme activity is highly temperature-dependent. When the temperature changes, the protein structure and vibration frequency of CA will change to different degrees, which will lead to changes in enzyme protein activity. In the natural environment, natural leaves can sustain their enzyme-based carbon sequestration capacity without requiring additional energy sources, utilizing natural light ($0-1 \times 10^5$ lux). Ecoleaf, much like natural leaves, has enzyme activity that is temperature-sensitive. As a result, this study employed spray staining (Supplementary Fig. 3a, b) to endow artificial leaves with a photosynthesis-like effect similar to that of their natural counterparts. This approach allows CA activity to be maintained by heat energy generated from light ($0-1 \times 10^5$ lux) under varying environmental conditions, without the use of an external heat source, in line with the natural photosynthesis process. As observed in Fig. 3a, b, when the leaf's ambient temperature was set at 30 °C, the initial warming rate and final temperature of the EcoLeaf increased as the K/S value increased, while the light level remained constant. Notably, the final temperature reached 55 °C when exposed to $9.4 \times 10^4$ lux light, surpassing the optimal carbon sequestration temperature of EcoLeaf (40 °C, Supplementary Fig. 3c). For practical $CO_2$ capture applications, the Ecoleaf variant with the most efficient photothermal conversion should be deployed in natural environments to achieve optimal carbon capture. In conditions of high visible light intensity, light avoidance methods (similar to greenhouses) can be employed to ensure CA's optimal carbon capture activity. Conversely, under low ambient temperature or weak light intensity, the light utilization efficiency of EcoLeaf can be adjusted to meet CA's optimal carbon capture temperature. This is achieved by either increasing the K/S value or incorporating materials with enhanced light-heat conversion capacity. This adaptability is essential for the broader climate applicability of artificial leaves. In this study, enzyme activities were also explored for two groups of EcoLeaf with different K/S values exposed to the same light level (Supplementary Fig. 3(d)). Under an initial temperature of 30 °C and after 3 min of irradiation with a light level of $4.5 \times 10^4$ Lux, enzyme activities for EcoLeaf with K/S values of 3.5 and 86.7 increased by 135%. This highlights EcoLeaf's capability to maintain its carbon sequestration features solely using visible light as an energy source, akin to real leaves, a critical aspect for sustainable development.

Light plays a pivotal role in governing the expansion and contraction of stomata on natural leaves. During daylight, in the presence of visible light, leaves tend to expand stomata to facilitate the transport of $CO_2$, oxygen, and water vapor, whereas at night, in darkness, they close stomata. This adaptive stomatal behavior is crucial for optimizing the cycle of photosynthesis and respiration. Drawing inspiration from this natural mechanism, we synthesized GMA/4,4'-AZO from GMA and 4,4'-AZO, capitalizing on azobenzene molecules' unique photoisomerization property, as depicted in Fig. 4a. Employing this principle, we grafted GMA/4,4'-AZO as a monomer to form a three-dimensional network matrix on the blade surface. When exposed to

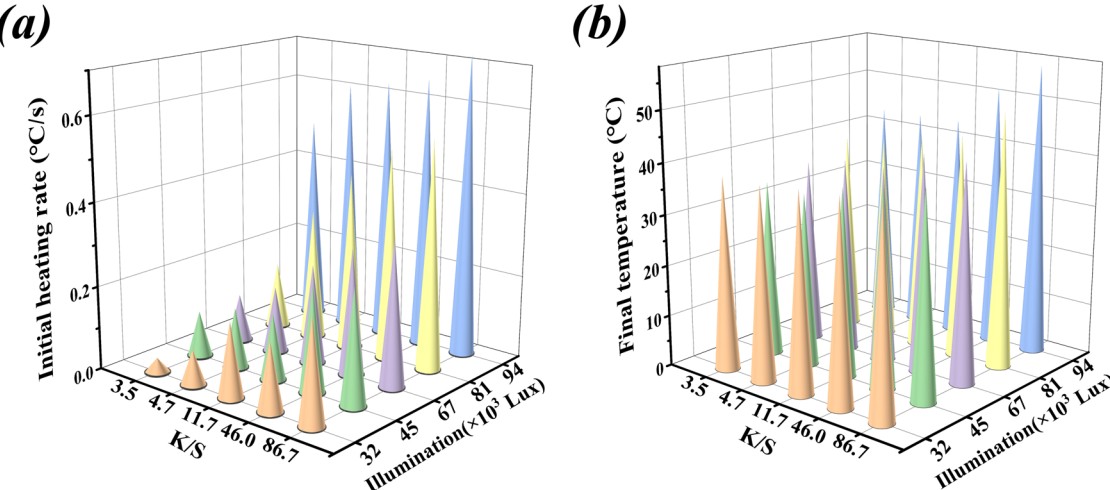

**Fig. 3 | Photothermal conversion performance of EcoLeaf.** Effects of K/S value and visible illuminance on (**a**) initial warming rate and (**b**) final arrival temperature of artificial leaves at an initial temperature of 30 °C.

various light sources, the matrix's mesh size cyclically and reversibly changes in response to the monomers' spatial orientation, thereby controlling the stomata of the artificial blade's expansion and contraction. Firstly, the stomata of EcoLeaf were visualized by high-resolution TEM in this study. As shown in Supplementary Fig. 4, with the increase in TEM magnification, the stomatal structure distributed on the 3D mesh mechanism can be clearly seen, and the stomatal size is only in the nanometer scale (0–2 nm). In addition, as illustrated in Fig. 4b, c, irradiation of EcoLeaf with a 450 nm visible light source (19.1 mW/cm²) and a 365 nm ultraviolet light source (20.3 mW/cm²) caused the absorption peaks of trans 4,4'-AZO at 390 nm to decrease and increase with irradiation time, respectively. This observation confirms that GMA/4,4'-AZO retains its photoresponsive properties, as shown in Supplementary Fig. 5, even after successful grafting onto the fiber-based material. The isomerization of GMA/4,4'-AZO leads to corresponding changes in the matrix's mesh size and pore volume, forming the foundation for light-driven pore size regulation. As depicted in Fig. 4d and Supplementary Fig. 6, one round of irradiation (450 nm, 365 nm, and 450 nm) resulted in decreased and increased then pore volume, surface aera and average pore size of EcoLeaf. This demonstrates that light excitation at different wavelengths can directionally regulate EcoLeaf's pore size characteristics, which, in turn, controls the $CO_2$ capture and the transport of $H_2O$ and carbon sequestration products. The pore size data further validate the range of pore sizes observed by high-resolution TEM. Additionally, we examined the variations in the optical contact angle of the EcoLeaf surface under cyclic excitation with different light wavelengths. As shown in Fig. 4e, light excitation at 450 nm decreased the water contact angle, while 365 nm excitation increased it. This change is primarily attributed to the alteration in stomatal aperture on the EcoLeaf surface. When irradiated with 450 nm light, the pore volume and average pore size increased, causing the water contact angle to decrease and vice versa. Thus, the light-responsive properties of the leaf's surface provide it with an adjustable stomatal structure. In practice, natural sunlight contains both 365 nm UV and 450 nm blue light. Therefore, the stomatal expansion and contraction of EcoLeaf under sunlight is critical. As shown in Supplementary Fig. 7, when exposed to natural sunlight containing both 450 nm and 365 nm, the EcoLeaf irradiated with the 365 nm wavelength is more inclined to absorb the blue light of 450 nm, which puts the stomata in a dilated state. Detailed analysis is located under Supplementary Fig. 7.

To explore the impact of stomatal expansion and contraction characteristics on carbonic anhydrase (CA), we measured CA's acid resistance, and high-temperature resistance using the esterase method

before and after light excitation at different wavelengths. As shown in Fig. 4f, g, CA exhibits better acid resistance and high-temperature resistance in the contracted-stomata state compared to the expanded-stomata state. This observation suggests that when the external conditions are suitable, expanded stomata enhance $CO_2$ transmission, increasing carbon sequestration *per* unit area. Conversely, when the stomata are constracted, the denser matrix mesh insulates CA from external substances, preserving enzyme activity and maintaining protein conformation[35,36]. This isolation is beneficial for resisting the challenges of natural environments, extending EcoLeaf's service life, and securing its functionality. In addition, during natural carbon sequestration, $CO_2$ enters the leaf body when the stomata are expanded. After the leaf stomata are constracted, $CO_2$ can be converted to high-value products through a stable dark reaction. Similar to this process, the stomatal expansion and contraction characteristics of EcoLeaf can sequester carbon when the stomata are expanded, and after the stomata are constracted, $CO_2$ can be converted into high-value products through stable and continuous catalytic conversion of $CO_2$ by a multi-enzyme cascade reaction.

The effect of stomatal expansion and contraction characteristics on the carbon capture performance of EcoLeaf was investigated in this study by the device shown in Fig. 5a. As depicted in Fig. 5b, the $CO_2$ capture rate of EcoLeaf was markedly reduced when exposed to 365 nm UV illumination, in contrast to the 450 nm. These changes can be attributed to the expansion and contraction of stomata. Expanded stomata offer larger pore sizes, surface area, and pore volume, facilitating substrate transportation to CA, EcoLeaf's primary carbon-sequestering component. Conversely, when the stomata contract, the leaf's pore size decreases, limiting the contact between $CO_2$ and CA to some extent and interrupting $CO_2$ sequestration. This control mechanism fine-tunes the $CO_2$ capture process by EcoLeaf. In addition, analyzing from the molecular point of view, the change in exposed groups due to the change in pore structure will also affect the contact probability of EcoLeaf with $CO_2$ to some extent. In summary, from the macroscopic changes in pore structure to the microscopic changes in the relative positions of molecules, the carbon fixation rate during stomatal contraction can be adequately explained to be lower than that during stomatal expansion.

The essence of photosynthesis in natural leaves lies in the process of carbon assimilation by the photosystem utilizing light energy. This process results in the synthesis of glucose from $CO_2$ and water, ultimately leading to the formation of starch stores[37,38]. In addition, water and carbon sequestration products share transport pathways. Water is primarily transported from the petiole to various parts of the plant

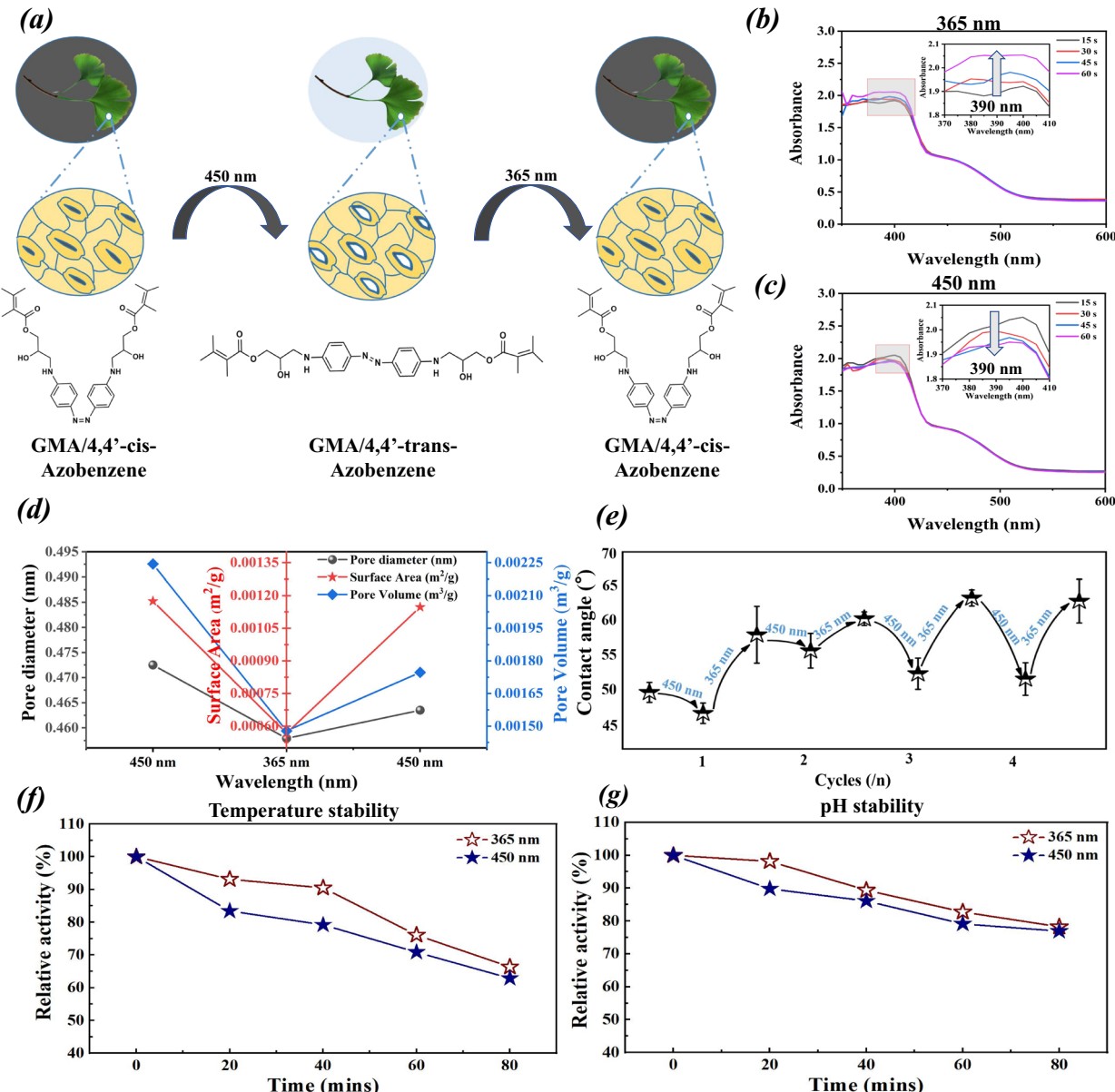

**Fig. 4 | Light response characterization and stability of EcoLeaf. a** Schematic diagrams of the changes in the molecular structure of GMA/4,4'-AZO and the stomata of EcoLeaf under different wavelengths of excitation light; photoresponse properties of EcoLeaf under (**b**) 365 nm UV and (**c**) 450 nm visible excitation; **d** Changes in the pore volume, average pore size and surface area of EcoLeaf under 450 nm visible and 365 nm UV excitation; **e** Changes in the optical contact angle of EcoLeaf under 450 nm visible and 365 nm UV excitation (Parallel experiments with three sets of identical samples, data are presented as mean values +/−SEM); Effects of stomatal expansion and contraction characteristics on the (**f**) temperature stability and (**g**) pH stability of EcoLeaf.

body through conduits, ultimately reaching the leaves, where it is utilized[39,40]. Carbon fixation products are transported through conduits to different parts of the plant, such as roots, stems, flowers, and fruits, to facilitate cell growth. This efficient capture of $CO_2$ by CA relies on the participation of $H_2O$. Additionally, if the carbonic acid produced by CA, after capturing $CO_2$, is retained within the artificial leaf, it can lower the pH in CA's vicinity, leading to the dissociation of enzyme subunits and affecting CA's $CO_2$ capture capacity. Hence, adequate water and carbon fixation product transport is crucial in EcoLeaf to maintain the activity of photosynthetic enzymes and prevent product inhibition. The structural properties of cellulose paper fibers are similar to those of natural leaf veins, allowing solutions to spontaneously diffuse and transport within them due to osmotic pressure. In this study, we examined the water-carbon sequestration product transport properties of EcoLeaf by immersing leaf petioles in branches

with ample water content (Fig. 1a). As evident in Fig. 5c, EcoLeaf, embedded with litmus, rapidly turned red upon absorbing gaseous $CO_2$. However, after placing its petiole in water for 20 min, the red litmus solution reverted to its original blue-violet color. This transformation occurs because the production of $H_2CO_3$ (carbonic acid) within EcoLeaf, following $CO_2$ capture, creates an acidic microenvironment, causing the litmus test solution to turn red. When the leaf petiole is immersed in water, $H_2CO_3$ within the EcoLeaf matrix migrates through the leaf's fibers to the aqueous solution, reducing the $H_2CO_3$ concentration and restoring the matrix's microenvironment to its initial state, turning the litmus test solution blue-purple again. Therefore, EcoLeaf's natural fiber mesh structure can serve as a conduit similar to those in natural leaves, facilitating the transport of water and $CO_2$ sequestration products. This capability opens the door to long-term cyclic carbon sequestration.

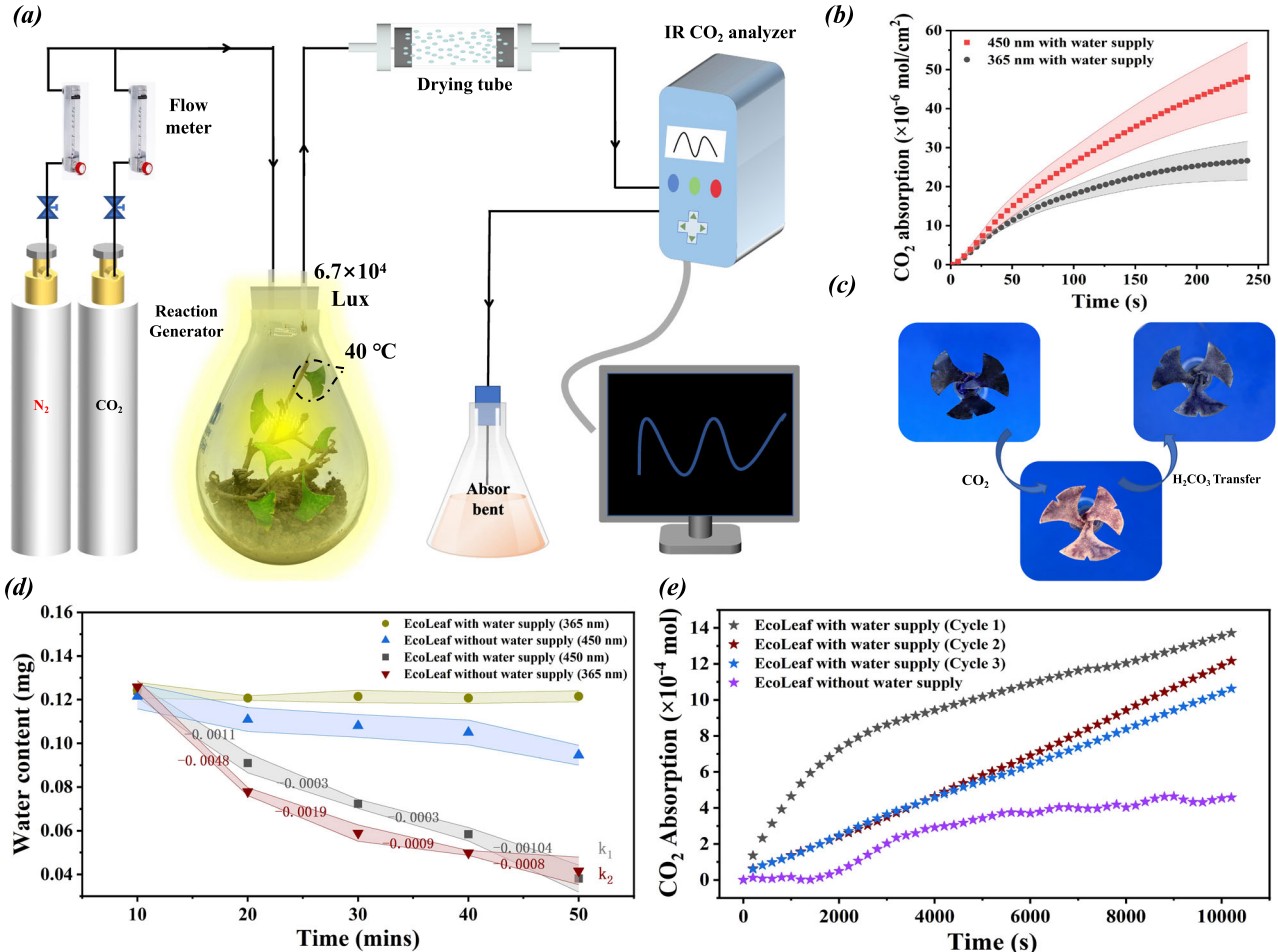

**Fig. 5 | Material transport characteristics and carbon sequestration of EcoLeaf.**
**a** Schematic diagram of the $CO_2$ capture device; **b** Effect of stomatal expansion and contraction states on the carbon capture performance of EcoLeaf (Parallel experiments with three sets of identical samples, data are presented as mean values +/−SEM); **c** Litmus color diagrams of EcoLeaf embedded with litmus test solution after $CO_2$ capture and litmus color diagrams of EcoLeaf after placing its petiole in deionized aqueous solution for transporting $H_2CO_3$ at a room temperature of 30 °C; **d** Changes in water content of EcoLeaf with and without water supply (365 nm and 450 nm) (Parallel experiments with three sets of identical samples); **e** Long-term carbon sequestration stability of EcoLeaf with and without water supply and cycling stability of EcoLeaf with water supply.

Artificial leaves, when exposed to the external environment for extended periods for gaseous $CO_2$ capture, experience surface water evaporation. This results in the loss of the proton source required for CA in capturing gaseous $CO_2$. Thus, timely replenishment of water is crucial. As illustrated in Fig. 5d, without a petiole connected to a water source, the water content in the EcoLeaf progressively diminishes over time, leading to water dissipation. Conversely, when the petiole is inserted into a watery branch, the leaf's water content remains relatively stable over time. This indicates that the water source within the branch continuously supplies water to the CA in the umbrella area under the influence of osmotic pressure. While the water content of the leaf blade remains constant, water and carbonic acid molecules are in constant motion. To further explore the impact of stomatal expansion and contraction characteristics on leaf water cycling properties, this study was conducted to examine the variations in water content of EcoLeaf under 365 nm and 450 nm light excitation, respectively. As shown in Fig. 5d, When the roots of EcoLeaf were in contact with a water source and the stomata were contracted (365 nm), a decrease in water content was observed compared to when the stomata were expanded (450 nm), indicating that the expansion and contraction states of the stomata have an impact on the water transport properties of EcoLeaf. With expanding stomata, driven by the osmotic pressure of the liquid, water molecules were more likely to

pass through the pores of the 3D mesh fabric matrix to reach the leaf surface, providing an adequate source of protons for CA carbon sequestration. However, when the stomata of EcoLeaf were contracted, a reduction in its own water content occurred, indicating a weakened rate of water transport. This weakening was primarily due to the stomatal contraction, which increased the density of the three-dimensional mesh fabric matrix of EcoLeaf and opposed the liquid osmotic pressure, hindering the transport of water molecules. This finding further supports the conclusion in the Fig. 5b. Furthermore, when there was no water supply, the rate of water loss ($k_2$) of EcoLeaf in the pore-contracting group (365 nm) was slightly higher than that of the pore-expanding group (450 nm, $k_1$) in the primary stage. This may due to the contraction process of the stomata induced by the UV light at 365 nm, which squeezed some of the water outside the EcoLeaf. Importantly, the decreasing rate of $k_2$ value was much lower than that of $k_1$. This is mainly attributed to the water retention effect caused by the contraction of the mesh, which prevents the volatilization of water molecules retained inside the mesh. As a result, the water content of the contracted group has become higher than the expanded group in the later stage. In conclusion, EcoLeaf can achieve a dynamic equilibrium, facilitating the water-carbonic acid transport process. This prevents the adverse effects of leaf dehydration and product inhibition on the carbon sequestration capacity of artificial leaves. Consequently,

EcoLeaf can consistently deliver its performance in capturing gaseous $CO_2$.

The long-term carbon sequestration stability of EcoLeaf is shown in Fig. 5e, $CO_2$ capture rate of EcoLeaf with water supply is significantly higher than that of EcoLeaf without water supply. This is mainly attributed to the greater proton availability in EcoLeaf with higher water content. This enhanced proton supply enhances CA's efficiency in converting $CO_2$, resulting in an overall improvement in EcoLeaf's carbon sequestration efficiency. The $CO_2$ capture ability of EcoLeaf with water supply also gradually stabilized over the long-term and cycling tests without significant product inhibition. In addition, the second and third batch sequestration rates were essentially unchanged over time, thus demonstrating that the sequestration behavior of EcoLeaf tends to be in transport equilibrium and does not result in inhibition of product accumulation. This is mainly attributed to its substance-transport properties. In addition, the $CO_2$ capture ability of EcoLeaf without water supply was unstable and gradually weakened due to the inhibition of products and proton sources. This further demonstrates that the material transport property enables the EcoLeaf prepared in this study to maintain a stable carbon sequestration capacity over a long period of time.

In order to investigate whether CA, the core organ of carbon sequestration, would undergo enzymatic leakage during $H_2O$-$H_2CO_3$ transport, the aqueous solution during transport was qualitatively and quantitatively analyzed by LSCM and Caumas Brilliant Blue staining (G250) in this study. As shown in Supplementary Fig. 8a, within 0−50 min after the petiole of EcoLeaf containing FITC-CA was connected to the water source, the aqueous solution contained only a very small amount of weak green fluorescent dots and did not show bright green fluorescence. This provides preliminary evidence that no leakage of CA occurred. In addition, Supplementary Fig. 8b shows that CA leakage rates remain extremely low, which further proves that no obvious enzyme leakage occurred in EcoLeaf. Additionally, in this study, the carbon capture core organ CA was replaced with formate dehydrogenase (FDH) to verify the designability of the EcoLeaf bionic system. As shown in Supplementary Fig. 9, $CO_2$ can be successfully converted to HCOOH by replacing CA with FDH and NADH in the EcoLeaf bionic system. This demonstrates the high degree of designability of the EcoLeaf system.

In order to further explore the practical application value of EcoLeaf, we have summarized emerging $CO_2$ conversion systems with high light energy conversion efficiency and analyzed material cost, preparation method, light utilization efficiency, and potential environmental impact (Supplementary Table 1). Compared to existing light source-driven $CO_2$ treatment systems, EcoLeaf has low-cost raw materials, mild preparation conditions, and is able to be degraded by natural soil and participate in further ecological cycles. The efficiency and cost of light energy utilization are decisive for the large-scale preparation and application of materials. Therefore, this study quantifies the prospect of large-scale application (valuation of large-scale application) of the materials through the efficiency/cost of light energy utilization. Supplementary Fig. 10 shows that the valuation of scale-up applications of EcoLeaf achieved 4.9 mol·$J^{-1}$·\$$^{-1}$ in the first place compared to existing carbon sequestration materials. It can be seen that EcoLeaf has significant advantages over existing carbon sequestration strategies after quantifying the economic costs and energy efficiency. Thus, the design and preparation of EcoLeaf provide a sustainable and effective way to achieve the goal of carbon neutrality.

Natural leaves ultimately wither and become part of the ecosystem's nutrient-cycling process. During this phase, leaves gradually degrade, releasing various elements that return to the soil. This nutrient source becomes available for the plant root system to absorb. However, the existing artificial leaf system cannot be degraded by the natural environment, and will face the problem of secondary pollution while sequestering carbon, which seriously limits its large-scale

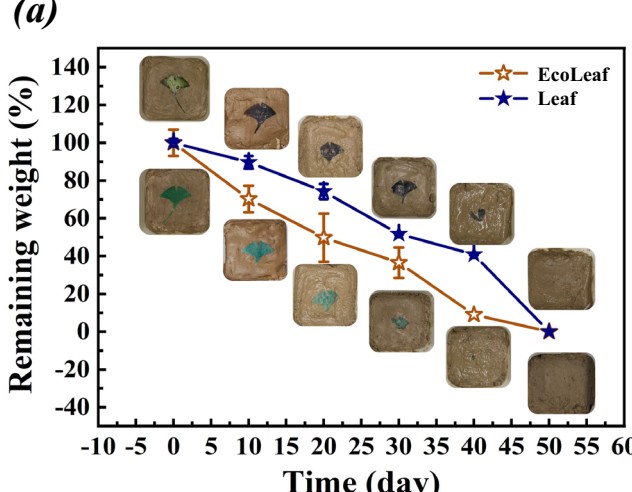

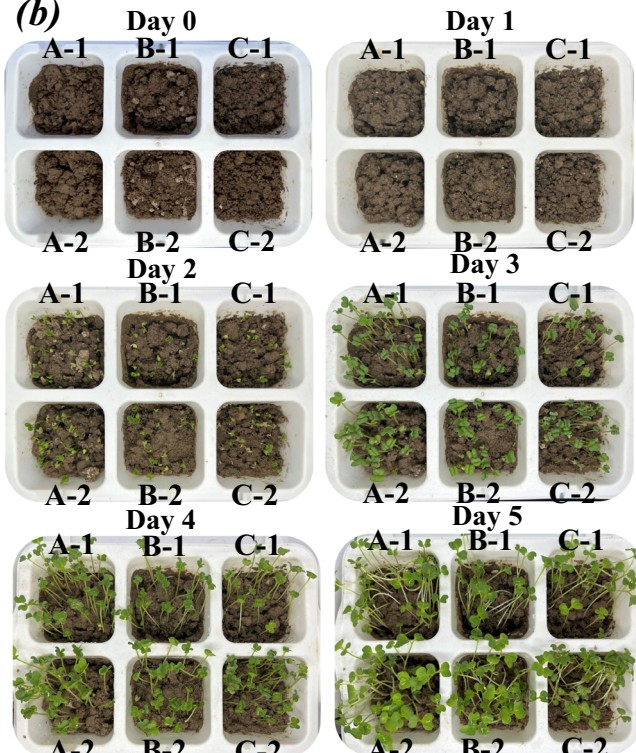

**Fig. 6 | Soil degradation characteristics of EcoLeaf. a** Soil degradation and mass loss of natural leaves and EcoLeaf at room temperature (Parallel experiments with three sets of identical samples, data are presented as mean values +/−SEM); **b** Soil eco-culture maps of natural leaves (A-1-A-2), EcoLeaf (B-1-B-2), and blank group (C-1-C-2) at room temperature.

application. Because EcoLeaf's primary framework closely resembles the cellulose fraction of natural leaves, it can effectively mimic the natural process of leaf degradation.

As depicted in Fig. 6a, natural leaves and EcoLeaf undergo the process of returning to the soil. The figure illustrates that EcoLeaf, similar to natural leaves, fully participates in the alternation between wet and dry soil conditions and the metabolic processes involving various microorganisms and enzymes. The degradation rate of EcoLeaf is slightly higher than that of natural leaves. By day 20, 50% of EcoLeaf had degraded in natural soil, while only 32% of natural leaves had degraded. By day 40, EcoLeaf had almost completely degraded, while natural leaves remained. This is primarily due to the fact that EcoLeaf's

main framework (cellulose) is more homogeneous compared to the composition of natural leaves, which include lignin, cellulose, and hemicellulose, making EcoLeaf more easily degradable.

To investigate whether the degradation products of EcoLeaf affect the soil microenvironment, we used the degraded soil to cultivate canola crops. As Fig. 6b demonstrates, soil from both EcoLeaf degradation and natural leaf degradation, as well as the canola seedlings planted in the control group, showed similar growth patterns. This is because the degradation products of both natural leaves and EcoLeaf consist of essential soil elements (C, H, O, and N) required for plant growth, and they do not disrupt the soil microenvironment. This indicates that EcoLeaf can smoothly participate in the ecosystem's nutrient cycling process, akin to natural leaves, illustrating the significance of the ancient Chinese poem, Turning into spring mud protects the flowers.

In this study, we successfully developed EcoLeaf, a biomimetic construct closely resembling natural leaves. The mechanical properties of EcoLeaf are higher than those of most natural leaves but are similar to those of palm leaves, making it less prone to pulverization and providing excellent environmental resistance. The EcoLeaf can convert visible light energy into controlled thermal energy to maintain the optimal activity of CA for carbon capture. This control is achieved by regulating the depth of leaf staining and light intensity. Through cyclic excitation using 365 nm and 450 nm light sources, the stomata within the EcoLeaf substrate cyclically contract and expand, allowing for the controlled switch between self-protection in harsh environments and high-efficiency carbon capture in suitable conditions. The connection of the petiole to a water source ensures a steady proton supply for capturing gaseous $CO_2$ while preventing the accumulation of carbon sequestration products. Soil degradation experiments indicate that EcoLeaf can completely degrade in natural soil within 40 days, and the resulting degradation products have no adverse effects on the soil microenvironment. This ensures a stable ecological cycle. Compared to existing synthetic blades, EcoLeaf offers the best efficiency/cost ratio and serves as a biomimetic and functionalized platform for artificial biocarbon sequestration. Moreover, its biocarbon sequestration pathway exhibits a certain degree of design flexibility, allowing for adaptation to various single-enzymatic or multi-enzymatic cascade catalysts for converting gaseous $CO_2$ into more abundant $C_2$/long-chain products. The EcoLeaf represents a significant step towards the development of enzymatic artificial leaves.

## Methods

### Chemicals
Cotton fiber paper (Homemade). Carbonic anhydrase (bCA, from *bovine erythrocytes*) purchased from Innochem (China). Formate dehydrogenase (FDH, from viruses) and Nicotinamide adenine dinucleotide (NADH, from viruses) purchased from Chongqing Ruiya Biotechnology Co. (China). Carboxylic acid (HCOOH), Isopropyl thioxanthone (ITX), Acetone, Glycidyl methacrylate (GMA), 4,4′-Azodianiline (4,4′-AZO), Tetrahydrofuran, *p*-Nitrophenol (*p*-NP), *p*-nitrophenylacetate (*p*-NPA), Brilliant Blue G (G-250), Fluorescein isothiocyanate (FITC) were purchased from Sigma-Aldrich Chemical Co. (USA). $CO_2/N_2$ (0.15/0.85 vol) standard gases were purchased from Shaanxi Fast Gas Co. (China). Other reagents used were analytical pure.

### Preparation process of EcoLeaf
Preparation of EcoLeaf substrates. The pulp evacuation revolution was 3000 r/min, the press pressure was 0.48 MPa, the press time was 2 min, the drying time was 8 min, and the drying temperature was 105 °C. The thickness and density of the EcoLeaf substrate were 0.22 mm and 0.61 g/cm³, respectively. Configure ink/ethanol solutions with concentrations of 0, 0.01, 0.1, 0.5, and 1(v/v). A gradient concentration of green ink (W-3, Five Thousand Years, China) was sprayed onto the surface of the leaf substrate (fiber-based paper) using a

spraying technique. The EcoLeaf was placed in a Petri dish, sealed, and subsequently placed in an electric blast drying oven (50 °C) for 3 h. This method resulted in the preparation of EcoLeaf with K/S values of 3.5, 4.7, 11.7, 46.0, and 86.7, respectively. The leaf shape was then outlined using a pencil, and the pattern was created by cutting.

Synthesis of photoresponsive monomer molecules. First, 26.4 μL of GMA (0.2 mmol) was dissolved in 100 mL of tetrahydrofuran solution along with 21.22 mg of 4,4′-AZO (0.1 mmol). Subsequently, 100 μL of triethylamine was added as a promoter. The reaction was heated and refluxed for 24 h at 60 °C, ensuring a complete reaction between the amino groups on both sides of the single 4,4′-AZO molecule and the epoxy groups of the two GMA molecules. The resulting solution was evaporated to a molten consistency, yielding a mixture of GMA, 4,4′-AZO, and GMA/4,4′-AZO. This mixture was dissolved in 100 mL of deionized water, filtered at 0 °C to remove insoluble matter, and then frozen and dried to obtain GMA or GMA/4,4′-AZO.

Grafting of photoresponsive mesh cloth substrates and CA immobilization. This process involved two main steps: (1) The photoinitiator isopropylthioxanthone dormant species (ITXSP) was applied to the surface of the leaf substrate (fiber-based paper) through a UV-induced extraction-hydrogen coupling reaction. (2) Under visible light irradiation, ITXSP generated surface radicals, initiating the polymerization of GMA/4,4′-AZO to create a three-dimensional reticulated matrix with light-responsive properties, encapsulating CA on the leaf substrate. The specific procedure was as follows (Fig. 1a): ITX/acetone saturated solution was uniformly coated on one side of the fiber-based paper, creating a sandwich structure between quartz plates. The structure was then irradiated with a UV-Hg lamp (365 nm, 0.245 W/cm²) for 6 min. The dormant species ITXSP was inoculated onto the surface of the fiber-based paper (Cp-ITXSP) to provide the site for grafting polymerization of GMA/4,4′-AZO. Subsequently, GMA/4,4′-AZO/CA(PBS) (1:1, v/v, 0.5 M, pH = 8 phosphate buffer) was applied to one side of the Cp-ITXSP. The sandwich structure was used to activate the dormant species of ITXSP under Xe lamp (>420 nm, 120 mW/cm², 1 h), initiating the polymerization of GMA/4,4′-AZO to create a three-dimensional mesh matrix and encapsulate CA within the pores. After grafting was completed, the surface of the artificial leaf was thoroughly cleaned using deionized water, completing the preparation of EcoLeaf. Detailed characterization and testing are shown in the supporting materials.

### Characterization and testing
**Structural characterization of EcoLeaf.** To characterize the successful synthesis of GMA/4,4′-AZO, the molecular structure and interactions of the substances before and after the reaction were analyzed using nuclear magnetic resonance spectroscopy (NMR, AVANCE NEO 600 MHz, BRUKER, Germany) with DMSO-$d_6$ as the solvent. The successful preparation of artificial leaves was confirmed by analyzing the changes in the apparent morphology of the fiber-based paper before and after grafting through scanning electron microscopy (SEM, TESCAN VEGA, TESCAN, Czech Republic). The distribution of immobilized CA on the fiber surface was observed using laser confocal fluorescence microscopy (LCSM, LSM800, CARL ZEISS AG, Germany). X-ray photoelectron spectroscopy (XPS, Thermo Scientific K-Alpha+, THERMO FISHER, America) was used to qualitatively analyze the elemental species and different chemical valence states of the atoms on the surface of the fiber-based paper at each stage of the grafting reaction. Visualization of stomata in 3D mesh substrates on EcoLeaf by transmission electron microscopy (TEM, FEI Talos F200X G2, Thermo Fisher Scientific, America).

### Mechanical properties of EcoLeaf
The mechanical properties of natural leaves from different species and EcoLeaf were investigated using the multifunctional testing machine. First, leaf samples were collected using a hand pressure sampler with a

standardized sampling width of 2 mm. This ensured consistency across the different specimens and minimized variations due to sampling methods. Next, the thickness of the leaf samples was measured using a precise thickness gauge. The measurements were recorded, providing important data for further analysis of the leaf mechanical properties. To ensure the samples remained securely in place during testing, they were fixed in the fixture of the EcoLeaf testing machine. This fixation step ensured that the samples would not slip or move during the subsequent tensile testing process. The leaf samples were then subjected to controlled stretching tests using the testing machine. The samples were stretched at a speed of 10 mm/min, allowing for the measurement of various data points such as force, displacement, and time. The thickness of the EcoLeaf 1 and EcoLeaf 2 used in this study was measured to be 0.19 mm and 0.22 mm, respectively, which is very similar to natural leaves. In addition, the densities of the two are 0.60 g/cm³ and 0.61 g/cm³ respectively. These collected data were then analyzed to assess the mechanical properties of the leaf samples.

## Light trapping by EcoLeaf

The photothermal conversion properties of the materials were evaluated using K/S values (K absorption coefficient, S scattering coefficient) of fiber-based papers with varying dye depths determined by employing a colorimeter (CI7800, X-rite, United States). The K/S values were measured by a photometer (1802, DELIXI, China) and a thermal imager (ST9660, CIMA, China) to evaluate the effect of light irradiance on the heating rate and final temperature of the EcoLeaf surface.

## Stomatal expansion and contraction characteristics of artificial leaves

The UV-visible spectrophotometer (UV-2600, Shimadzu, China) was employed to investigate the UV-absorption spectra of an aqueous solution of the photoresponsive monomers GMA/4,4'-AZO (1 μL/20 mL) and EcoLeaf under visible light at 450 nm and UV illumination at 365 nm. Additionally, a BET-physical adsorbent meter-ratio surface area analyzer (ASAP 2460, Micromeritics, United States, Analysis adsorptive: $CO_2$, Analysis bath temp: 273 K) and a video-optical contact angle measuring instrument were used to examine changes in pore size characteristics and the optical contact angle of EcoLeaf before and after excitation with 365 nm UV light and 450 nm visible light (10 min).

## Water-product dynamic transport characterization and capture performance of EcoLeaf for gaseous $CO_2$

To investigate the water transport characteristics of EcoLeaf, the petioles of the EcoLeaf were immersed in water and air. The weight of the leaves was recorded every 10 min, and the water content was calculated. The carbonate transport properties of the product were examined using the litmus colorimetric method. Initially, the litmus test solution was mixed with the GMA/4,4'-AZO grafting system at a 1:1 (v/v) ratio. Subsequently, litmus molecules were embedded within the three-dimensional mesh matrix through the visible-light-induced graft polymerization technique, as described earlier. After grafting, any unreacted litmus test solution and GMA/4,4'-AZO were removed by soaking in deionized water. Finally, the product was placed in a sealed bag, and $CO_2$ gas was passed through it. Changes in the color of the EcoLeaf surface were observed and recorded. Additionally, petioles of the EcoLeaf were placed in deionized water, and any color change in the EcoLeaf surface was recorded after 20 min.

To investigate the capacity of EcoLeaf to capture gaseous $CO_2$, a 10 cm² section of EcoLeaf was inserted into a water-containing hollow tree branch and left it for 10, 30, and 50 min, respectively, and then removed the EcoLeaf and placed it in a round-bottomed flask. The capture ability of EcoLeaf for gaseous $CO_2$ was examined under natural sunlight ($6.7 \times 10^4$ Lux). The initial temperature of the leaf surface was

30 °C, and the EcoLeaf body temperature was finally maintained at 40 °C by $6.7 \times 10^4$ Lux light. The gas used in these experiments was $CO_2/N_2$ (0.15/0.85 vol) standard gas. The gas flow rate at the inlet was controlled at 200 mL/min using a gas flow meter. The volumetric concentration of $CO_2$ in the outlet gas was determined with a $CO_2$ infrared analyzer (AW-T6, Zhaowei, China), and the $CO_2$ capture efficiency of the reaction system was calculated using Eq. (1).

$$CO_2 \ absorption \ rate = Q_{inlet}y_{CO_2,inlet} - Q_{outlet}y_{CO_2,outlet} \quad (1)$$

$Q_{inlet}$ and $Q_{outlet}$ denote the inlet and outlet gas molar flow rates (mol/min), respectively, calculated from the gas volume flow rate (mL/min) and gas concentration (mol/mL). $y_{CO_2,inlet}$ and $y_{CO_2,outlet}$ denote the gas inlet and outlet $CO_2$ mole fractions, respectively.

## Composition analyses of EcoLeaf

The pure NADH, NAD⁺, $H_2CO_3$, FDH, HCOOH dissolved in deionized water and the generating solution of the EcoLeaf containing FDH, NADH were determined using high-performance liquid chromatography (HPLC) (Shimadzu, Japan) with an Aminex HPX-87H column (300 × 7.8 mm², America) at 65 °C with a refractive index detector(RID-20A) at 40 °C, and 0.005 M $H_2SO_4$ at 0.5 mL/min was used in the mobile phase.

## Soil degradability and degradation products of EcoLeaf and natural leaves tested for eco-farming ability

The biodegradability of EcoLeaf and natural leaves was evaluated in natural soil. Both were buried in the soil at a depth of 5 cm and left in an outdoor environment for natural degradation. Periodically, the EcoLeaf and natural leaves were thoroughly washed with deionized water, dried at 40 °C, and then weighed and photographed for documentation.

In order to investigate whether the degradation of EcoLeaf affects the ecological tillage capacity of natural soils, 20 canola seeds were sown in soils in which natural leaves, EcoLeaf degraded leaves, and external natural environmental soils were placed. The ecological capacity of the soil was evaluated by observing and recording the growth of the oleaginous seeds within 5 days.

## Enzyme loading assay of EcoLeaf[41,42]

A gradient of CA/PBS solution (0.05–0.12 mg/mL, pH = 8) was placed in a 2 mL centrifuge tube, and 990 μL of G-250 rapid staining solution was added and allowed to stand in the dark for 2 min. The absorbance of the enzyme solutions at different concentrations was measured by a UV-visible spectrophotometer (UV-2600, Shimadzu, China). The concentration-absorbance standard curve was plotted. The absorbance of 100 μL of the remaining enzyme solution from the CA immobilization system was measured in a centrifuge tube, and the above procedure was repeated. The enzyme concentration and enzyme amount of the remaining enzyme solution were calculated from the standard curve, and the enzyme loading was obtained after making the difference.

## Enzyme activity assay of EcoLeaf[43,44]

The activity of CA was tested using the esterase method. The absorbance of p-NP/PBS buffer (pH = 8) with a concentration gradient of 0.01–0.045 mM was measured at 400 nm using a UV-visible spectrophotometer, and a concentration-absorbance standard curve was plotted. EcoLeaf and free CA were placed in 2 mL of PBS buffer, and 100 μL of p-NPA/acetonitrile (0.01 M) was added. The absorbance of the solution system at 400 nm was measured after the reaction system was reacted for 3 min under different intensities of visible light. The concentration of the product, p-NP, was calculated by a standard curve. In order to eliminate the effect of self-decomposition of p-NPA, a blank control experiment was performed in this study, in which the

concentration of the hydrolysis product *p*-NP was measured in buffer without enzyme, and the difference was made to obtain the amount of substrate decomposition actually catalyzed by CA. One unit of enzyme activity was expressed as the amount of enzyme required to release 1 μmol of *p*-NP per minute. The relative catalytic performance of immobilized CA under different conditions was characterized by relative activity (Eq. (2)).

$$Relative\ activity(\%) = \frac{Enzyme\ activity\ of\ each\ condition}{Enzyme\ activity\ of\ optimum\ condition} \times 100\%$$

(2)

### Determination of optimum carbon fixation temperature of EcoLeaf

The reaction system containing only 2 mL of PBS buffer (0.5 M, pH = 8) was placed at 10–50 °C for 5 min for constant temperature, and Eco-Leaf and free CA were added separately. The relative catalytic activity (Relative activity, Eq. (2)) of the artificial leaves and free CA were determined by the esterase method at different temperatures, respectively.

### Reporting summary

Further information on research design is available in the Nature Portfolio Reporting Summary linked to this article.

## Data availability

Source data are provided with the paper. All data underlying this study are available from the corresponding author upon request. Source data are provided with this paper.

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

## Acknowledgements

This work was supported by National Natural Science Foundation of China (52103008 (X.Z.)), 52003146 (B.H.)); Shaanxi Provincial Key R&D Program (2024SF-YBXM-607 (X.Z.)); Key Projects of Shaanxi Provincial Department of Education (23JY013 (X.Z.)); Key Program for International S&T Cooperation Projects of Shaanxi Province, China (2023-GHZD-14 (B.H.)).

## Author contributions

Xing Zhu: Investigation, Review & editing, Funding acquisition; Chenxi Du: Investigation, Methodology, Writing-original draft; Bo Gao: Writing-review & editing; Bin He: Supervision, Funding acquisition, Writing-review & editing.

## Competing interests

The authors declare no competing interests.
