## [Peer Review File · Nature Communications]

Reviewers' Comments:

Reviewer #1:

Remarks to the Author:

The manuscript outlines a strategy for fabricating an artificial leaf, EcoLeaf, capable of converting visible light into energy to accelerate CO₂ conversion in embedded enzymes. The introduction of azobenene in the material matrix enables EcoLeaf to mimic stoma open and closure, similar to a natural leaf. However, the manuscript lacks critical characterizations and falls short of demonstrating sufficient novelty for consideration in Nature Communications. Below are major and minor comments to enhance the manuscript:

Major Issues:

1. The authors mention that the paper substrate used to fabricate EcoLeaf shares fundamental properties with a natural leaf, including mechanical properties. However, there is no data supporting this claim, and specific details on materials, density, and thickness are lacking. The statement only mentions "cotton fiber paper (homemade)," and the material properties need to be characterized and explained in detail.
2. One of the key novelties the authors claim is the mimicry of stoma open and close, influencing CO₂ absorption and water transportation. However, no data in the manuscript supports this claim. The authors should incorporate data, especially in Fig. 4, demonstrating how changes in pore size affect these crucial properties of EcoLeaf. This can include showcasing the control of pore size to adjust CO₂ absorption rates and water adjustments.
3. The authors state the highest carbonic anhydrase (CA) activity in their system is at 40°C. However, in Fig. 4, where the experiment under natural sunlight is mentioned, the manuscript lacks details on temperature and intensity. This inconsistency needs clarification, and the information should be included in the Fig. 4a schematic.
4. A comparative analysis of light energy usage or CO₂ conversion performance with state-of-the-art systems is recommended to establish the significance of the proposed system.
5. There is a lack of detailed schematics illustrating the procedure for making EcoLeaf. While line 117 refers to Fig. 1a for the specific procedure, Fig. 1a does not present key fabrication processes. Enhancing the schematic to highlight the novelty of the materials/systems is suggested.
6. Pore size analysis, usually conducted through nitrogen absorption at 77K, is not detailed in the manuscript. Pore size distribution curves should be provided, and the apparent lack of a decrease in pore volume in Fig. 3d needs to be addressed.

Minor Issues:

1. The manuscript lacks organization, with two experimental sections and repeated paragraphs. Streamlining the structure for better readability is recommended.
2. Vague terms exist throughout the experiment section. For instance, clarification is needed on the UV wavelength used and its intensity (line 119), as well as definitions for terms like C-ITXSP (line 120) and CA(PBS) (line 121).
3. The pores shown in Fig. 1b-e are on a 10-100 μm scale, while those opened and closed by azobenzene are in the tens of nm. The authors should clarify how these different sizes of pores are formed. SEM or TEM images of the pores, especially before and after closure, are suggested.
4. An explanation is needed for the observed decrease in free CA activity when the temperature exceeds 40°C.

Addressing these comments will significantly enhance the manuscript's quality and potential for

publication.

Reviewer #2:

Remarks to the Author:

The work presented in the manuscript entitled "Leaves from painting to reality" is intended to show the engineering of an artificial leaf (named "EcoLeaf") for CO₂ sequestration from the atmosphere, mimicking the function of plant leaves.

The concept of engineering synthetic materials able to convert atmospheric CO₂ into valuable chemicals is keynote. Differently from the majority of previous works based on metal catalysts, or semiconductors, the authors engineered an artificial leaf where the activity of the material depends on the enzyme carbonic anhydrase (CA).

EcoLeaf is able to harvest light and convert it into thermal energy for CA activity (1), regulate CO₂ sequestration rate by opening-closing pores into a light-responsive polymeric matrix (2), transport the product of the enzymatic reaction outside the leaf (3), and finally degrade into the environment (4).

The experimental work has been detailed precisely, especially the synthesis and characterization of the light-responsive 3D polymer matrix embedding the CA enzyme.

Small technical points could be improved such as in line 98 ("3000 rpm"), citing Figure 4(a) in the main text of the manuscript (to the best of my reading I haven't found it), as well as better describing the light-trapping varying dye depths in the section "experiment".

Nevertheless, mimicking Nature does not necessarily mean attempting to reproduce artificially what Nature has already built; on the contrary it means getting inspired by Nature for developing/engineering solutions for the needs we want to tackle. For this reason, I think that the shaping of EcoLeaf as a plant leaf (lines 69 and 102) justifies the title of the manuscript ("Leaves from painting to reality") but it is not of key importance, as any surface able to harvest light, regulate CO₂ sequestration and transport, as well biodegrade could serve the purpose, as well as be more easily integrated with other materials for building complex bio-inspired composite materials.

Moreover, in the text of the manuscript (e.g. line 314: "...interrupting photosynthesis", or line 346: "...photosynthesis products.") the concept of "photosynthesis" is introduced.

In Nature photosynthesis converts inorganic carbon (CO₂) into organic compounds ("synthesis") by using light ("photo") as a source of energy to fuel the Calvin Benson-Bassham (CBB) cycle. Furthermore, in the introduction, the authors mentioned the work performed by Erb and coworkers who developed a new-to-nature synthetic pathway (Schwander et al. Science, 2016) for fixing CO₂ in vitro, i.e. "synthesis", and successfully achieved to power the cycle by light, that is "photo", in chloroplast mimics (Miller et al. Science, 2020) so to perform "photo" + "synthesis" that is "photosynthesis" in vitro.

Despite the fact that the interconversion of CO₂ and HCO₃⁻ catalyzed by the CA enzyme is fundamental in Nature to maintain the biological flux that takes place via CO₂ fixation by Rubisco, the study described in this manuscript lacks any further enzymatic step to convert CO₂ into C_n organic compounds, that is it lacks "synthesis". Therefore, I think that "CO₂ sequestration" should be used instead of "photosynthesis".

On the other hand, EcoLeaf is a first step towards the engineering of enzymatic artificial leaves, so I hope that this study could open doors for the engineering of "truly photosynthetic" artificial leaves in the future.

Reviewer #3:

Remarks to the Author:

The paper entitled "Leaves from Painting to Reality" by Zhu et al. reported an artificial leaf with

adjustable stomata for carbon capture. The authors further highlighted that Ecoleaf is easy to degrade in the ecosystem, making it a highly attractive green technology. However, it is not clear how the mimicking of the opening/closing of the stomata of artificial leaf is related to the performance of CO₂ sequestration from the atmosphere. In natural leaves, the stomata are closed to prevent excessive loss of moisture and opened for photosynthesis when water and sunlight are abundant. If the proposed artificial leaf is for CO₂ sequestration, there seems to be no reason for the closure of artificial stomata when under relative lower intensity of light irradiation. Furthermore, no evidence is presented to show the scalability and long-term cyclic performance of the artificial leaves. For example, the subsequent transportation of the HCO₃ generated in the Ecoleaf needs to be considered for practical applications.

The detailed comments are as follows:

- (1) What's the main purpose of the adjustable stomata in the Ecoleaf for improving the efficiency of carbon capture? Is there a link between the adjustable stomata and the efficiency of carbon capture? The author declares the CA would be stable under the closing state, but how do we adjust that during the practical application (365 nm)?
- (2) Cyclic and long-term testing of the capability to CO₂ capture are not mentioned.
- (3) For the LCSM, please give the results for the fluorescence intensity of FITC with the depth. It is very important to prove the distribution of the CA in the Eco-leaf.
- (4) Figure 3f has three diagrams. They should be labelled in 3(f), 3(g) and 3(h). Why there are two bars (results) for 365nm in 3f? It appears that the differences the temperature stability and pH stability under 365 and 450 nm irradiation are not so large. Are they really significant?
- (5) The sunlight contains both 365 and 450 nm light, so how sunlight regulates the opening and closing of stomata on the Ecoleaf needs to be clearly explained.
- (6) More measurements of the capture performance of the Ecoleaf need to be provided. The results in Figure 4d just show the short-time absorption of the CO₂.
- (7) What's the difference in Ecoleaf absorption of CO₂ performance under different light irradiation?
- (8) Considering the low solubility and stability of H₂CO₃, is it effective to migrate the H₂CO₃ from the Ecoleaf to the bulk water for long-term operation? Repeated testing is required.
- (9) Was the enzyme transferred to the bulk solution with the H₂CO₃ during the regeneration process?
- (10) Please provide more characterizations of the Eco-leaf, such as its surface pores structure under high-resolution SEM, mechanical strength, etc.

REVIEWER COMMENTS

Reviewer #1 (Remarks to the Author):

The manuscript outlines a strategy for fabricating an artificial leaf, EcoLeaf, capable of converting visible light into energy to accelerate CO₂ conversion in embedded enzymes. The introduction of azobenene in the material matrix enables EcoLeaf to mimic stoma open and closure, similar to a natural leaf. However, the manuscript lacks critical characterizations and falls short of demonstrating sufficient novelty for consideration in Nature Communications. Below are major and minor comments to enhance the manuscript. Addressing these comments will significantly enhance the manuscript's quality and potential for publication.

Response

Thank you for taking the time to review our manuscript. We appreciate your constructive comments and have carefully considered each one to improve the paper. In the revised manuscript, we have added key characterizations to enhance the depth of the paper and further enhance the innovation of the article.

Major Issues:

1. The authors mention that the paper substrate used to fabricate EcoLeaf shares fundamental properties with a natural leaf, including mechanical properties. However, there is no data supporting this claim, and specific details on materials, density, and thickness are lacking. The statement only mentions "cotton fiber paper (homemade)," and the material properties need to be characterized and explained in detail.

Response 1

Thank you for your comments. We understand that this information is crucial for a comprehensive understanding of our research. We have included detailed information on the preparing of the cotton fiber paper, including the density and thickness measurements in the revised manuscript (Supporting Information P1 L24-33). The density (0.60 g/cm³ and 0.61 g/cm³) and thickness (0.19 mm and 0.22 mm) of Ecoleaves were found to be similar to natural leaves. Additionally, we conducted stress-strain tests on five natural blades from different sources and two EcoLeaves using a multifunctional testing machine to provide data to support our mechanical performance requirements. The test results are shown in Figure 2. Comprising lignocellulose, the mechanical properties of the paper used to make EcoLeaf are higher than those of most natural leaves but

similar to palm fronds. The detailed experimental procedure are shown in Supporting Information (P3 L75-90) and result analysis are shown in the main text (P7 L158-177). Thank you for bringing this to our attention.

Figure 2. (a) Natural leaves from *Trachycarpus fortunei*, *Cercis chinensis bunge*, *Pyrus ussuriensis*, *Zamioculcas zamiifolia engl*, *Holly*, and *EcoLeaf* respectively; (b) Stress-strain curves and (c) Young's modulus of natural leaves from different sources and *EcoLeaf*

2. One of the key novelties the authors claim is the mimicry of stoma open and close, influencing CO₂ absorption and water transportation. However, no data in the manuscript supports this claim. The authors should incorporate data, especially in Fig. 4, demonstrating how changes in pore size affect these crucial properties of *EcoLeaf*. This can include showcasing the control of pore size to adjust CO₂ absorption rates and water adjustments.

Response 2

Thank you for your insightful feedback. In order to further investigate the effects of stomatal contraction and expansion on the CO₂ capture and water transport properties of *EcoLeaf*, we added two crucial characterizations. First, as depicted in Figure 5(b), the CO₂ capture rate of *EcoLeaf* was markedly reduced when exposed to 365 nm UV illumination, in contrast to the 450 nm. This phenomenon may be due to two reasons: first, it is attributed to the contraction of the mesh pores within the 3D fabric matrix under UV light excitation (Figure 4(d-e)), which obstructs the contact pathway between gaseous CO₂ and the core carbon-fixing enzyme CA; In addition, the redistribution of hydrophilic and hydrophobic groups under different wavelengths of light may also affects the CO₂ capture efficiency of *EcoLeaf*. Consequently, the carbon capture rate of

EcoLeaf decreased significantly. Detailed analysis can be found in the main text (P11 L275-286).

Figure 5. (b) Effect of stomatal contraction and expansion states on the carbon capture performance of EcoLeaf

Furthermore, our previous studies have revealed that, apart from stomatal contraction traits, the water transport characteristics also have a significant impact on the carbon sequestration performance of CA. Therefore, we investigated the influence of stomatal expansion and contraction traits on the water transport properties of EcoLeaf. As shown in Figure 5(d), When the roots of EcoLeaf were in contact with a water source and the stomata were contracted (365 nm), a decrease in water content was observed compared to when the stomata were expanded (450 nm), indicating that the expansion and contraction states of the stomata have an impact on the water transport properties of EcoLeaf. With expansive stomata, driven by the osmotic pressure of the liquid, water molecules were more likely to pass through the pores of the 3D mesh fabric matrix to reach the leaf surface, providing an adequate source of protons for CA carbon sequestration. However, when the stomata of EcoLeaf were contracted, a reduction in its own water content occurred. This weakening was primarily due to the stomatal contraction, which increased the density of the three-dimensional mesh fabric matrix of EcoLeaf and opposed the liquid osmotic pressure, hindering the transport of water molecules. This result also suggests that when the stomata are in a contracted state, water mass transfer has a greater effect on leaf water content than water evaporation. As a result, this process disrupted the catalytic conversion reaction of CA with CO₂ by reducing the number of protons while maintaining CA stability. This finding further supports the conclusion in the Figure 5(b).

Figure 5. (d) Changes in water content of EcoLeaf with and without water supply (365 nm and 450 nm)

3. The authors state the highest carbonic anhydrase (CA) activity in their system is at 40 °C. However, in Fig. 4, where the experiment under natural sunlight is mentioned, the manuscript lacks details on temperature and intensity. This inconsistency needs clarification, and the information should be included in the Fig. 4a schematic.

Response 3

Thank you for bringing up this inconsistency and providing valuable feedback. We apologize for not providing specific details on the temperature and intensity of natural sunlight in Fig. 4. In the revised manuscript, we have provided detailed labeling of the light intensity (6.7×10^4 Lux) and surface temperature (40 °C) of the EcoLeaf in Figure 5(a) (P13 L313-314). Additionally, we have refined the relevant experimental procedures in the experimental section (Supporting Information P5 L122-125). This clarification will ensure the accurate representation of our findings and eliminate any confusion caused by the lack of information.

4. A comparative analysis of light energy usage or CO₂ conversion performance with state-of-the-art systems is recommended to establish the significance of the proposed system.

Response 4

Thank you for the suggestion. We agree that a comparative analysis with state-of-the-art systems would provide valuable insights into the significance of our proposed system. In order to further explore the practical application value of EcoLeaf, we have summarized emerging CO₂ conversion systems with high light energy conversion efficiency and analyzed material cost, preparation method, light utilization efficiency, and potential environmental impact in the revised manuscript

(Table S1). Compared to existing light source-driven CO₂ treatment systems, EcoLeaf has low-cost raw materials, mild preparation conditions, and is able to be degraded by natural soil and participate in further ecological cycles. The efficiency and cost of light energy utilization are decisive for the large-scale preparation and application of materials. Therefore, this study quantifies the prospect of large-scale application (valuation of large-scale application) of the materials through the efficiency/cost of light energy utilization. Figure S10 shows that the valuation of scale-up applications of EcoLeaf achieved 4.9 mol·J⁻¹·\$⁻¹ in the first place compared to existing carbon sequestration materials. Thus, the design and preparation of EcoLeaf provide a sustainable and effective way to achieve the goal of carbon neutrality. This analysis helped us establish the unique contributions and potential advantages of our proposed system (P15 L376-389).

Figure S10. The prospect of large-scale application (valuation of large-scale application) of the materials through the efficiency/cost of light energy utilization (mol·J⁻¹·\$⁻¹) of different photofunctional CO₂ conversion systems

Table S1 Emerging CO₂ conversion systems with high light energy conversion efficiency

Raw materials	Synthesis method and reaction conditions	Light energy usage and CO ₂ conversion performance(mol·J ⁻¹ ·g ⁻¹)	Cost (\$/g)	Potential environmental impacts	Ref
EcoLeaf	Visible light induced graft polymerization	17.3	3.5	Soil degradable non-toxicity	-
Mn-MIL-88A	Hydrothermal treatment (65 °C/12 h)	0.14	-	Airborne particulate matter	1
g-C ₃ N ₄	One-pot thermal reaction (550 °C/4 h)	0.2	3116.7	Excessive energy consumption Increased greenhouse effect	2
Zn-MIL-88A	Hydrothermal treatment (65 °C/12 h)	0.36	-	Airborne particulate matter	1
PVK FDH	Spin coating	0.44	-	-	3
Cu ₂₇ Pd ₇₃ PVK Cu ₉₁ In ₉	and annealing (300 °C)	0.76	-	Heavy metal pollution	3
PVK CoPL		0.95	-	Water pollution	3
Cs ₃ Sb ₂ I ₉	Vacuum in-situ crystallization (120 °C/20 min)	1.67	9.8	Water and soil pollution	4
CsPbBr ₃ /BP	LARP; Self-assembly(-)	2.32	39.9	Heavy metal pollution	5
CsPbBr ₃ /USGO/a-	Hot injection and	2.66	-	Heavy metal	6

Fe ₂ O ₃	Self-assembly (180 °C/12 h)				pollution	
CsPbBr ₃	Facet manipulation (120 °C/1h, 220°C/15 h)	2.76	20.9	Heavy metal	pollution	7
CsPbBr ₃ /CTF-1-N i	Hot injection (120 °C/1.5 h, 100 °C/24 h); Self-assembly	3.11	3.3	Heavy metal	pollution	8
Ni-doped CsPbBr ₃ /Bi ₃ O ₄ Br	Hot injection (100 °C/1.5 h); Self-assembly	4.14	23.3	Heavy metal	pollution	9
T-SrTiO ₃ /CsPbBr ₃	Hot injection (-); In-situ growth	4.33	4.7	Heavy metal	pollution	10
FAPbBr ₃ /Ti ₃ C ₂	Hot injection (-); Interfacial interaction	4.39	3.0	Heavy metal	pollution	11
Co _{1%} @CsPbBr ₃ /C s ₄ PbBr ₆	LARP (-)	4.4	2.9	Heavy metal	pollution	12
Co/Co–Al ₂ O ₃	Calcine (150 °C/24 h, 500 °C/8 h, 700 °C/1 h)	4.42	1.5	Ecotoxicity	Excessive energy consumption Increased greenhouse effect	13
Cs ₃ Bi ₂ Br ₉	Hot injection (-)	4.59	-	Soil and water	pollution	14

Au rod and copper-palladium (CuPd) alloy shell	Epitaxial growth method (-)	4.95	226.9	Heavy metal pollution	15
CsPbBr ₃ -SOBr ₂ /g-C ₃ N ₄	Hot injection (120 °C/30 min, 150 °C/1h, 500 °C/2 h); Self-assembly	6.84	37.1	Water pollution	16
Pd@Nb ₂ O ₅	Microwave-assisted reaction (150 °C/ 20 min)	7.2	194.4	Heavy metal pollution	17
CsPbBr ₃ -Ni(tpy)	Hot injection (100 °C/1 h); Ligand exchange; Self-assembly	15.52	3.3	Water pollution	18
CsPbBr ₃ /PbSe	Hot injection (100 °C/1 h); Self-assembly	19.97	8.2	Soil and water pollution	19
Ternary g-C ₃ N ₄ /TiO ₂ /Ti ₃ Al C ₂ 2D/0D/2D composite	Sol-gel method (500 °C/4h, 550 °C/4h)	75.73	19.1	Microplastic pollution Excessive energy consumption Increased greenhouse effect	20
Rh/Al nanoantenna	One-pot thermal reaction	175.22	1634.6	Soil and water pollution	21

5. There is a lack of detailed schematics illustrating the procedure for making EcoLeaf. While line 117 refers to Fig. 1a for the specific procedure, Fig. 1a does not present key fabrication processes. Enhancing the schematic to highlight the novelty of the materials/systems is suggested.

Response 5

Thank you for your feedback. In the revised manuscript, we have added detailed preparation process of EcoLeaf in Figure 1 to highlight the novelty of the materials and system (P5 L127). Detailed experimental procedures are located at Supporting Information (P1 L23-59). Thank you for bringing this to our attention.

Figure 1. (a) Construction and performance of EcoLeaf; (b) The detailed preparation process of EcoLeaf

6. Pore size analysis, usually conducted through nitrogen absorption at 77K, is not detailed in the manuscript. Pore size distribution curves should be provided, and the apparent lack of a decrease

in pore volume in Fig. 3d needs to be addressed.

Response 6

Thank you for your feedback. First, in the revised manuscript, we present experimental parameters regarding the BET test (Supporting Information P4 L103). In addition, in order to obtain more rigorous experimental conclusions, we re-performed the sample preparation and changed the N₂ gas source to a CO₂ gas source to explore the pore properties of the samples. As shown in Figure 4(d), the average pore size, surface area, and pore volume of EcoLeaf under the irradiation of light cycles of 450 nm, 365 nm, and 450 nm undergo a trend of decreasing and then increasing (P10 L215-217). The corresponding pore size distribution curves have been added to the supporting information (P9 L216-218). In addition, the problem regarding incomplete recovery of the pore volume may be due to the stacking of Ecoleaf sample during the BET test. A portion of the EcoLeaf sample can not be able to receive sufficient light to restore the initial mesh structure.

Minor Issues:

7. The manuscript lacks organization, with two experimental sections and repeated paragraphs. Streamlining the structure for better readability is recommended.

Response 7

Thank you for your feedback. After referring to the format of several recent articles in Nature Communications, we have reorganized the article structure and removed some redundant sentences and paragraphs to enhance the readability of the article. In addition, we have combined the two experimental sections into one to improve the readability of the article.

8. Vague terms exist throughout the experiment section. For instance, clarification is needed on the UV wavelength used and its intensity (line 119), as well as definitions for terms like C-ITXSP (line 120) and CA(PBS) (line 121).

Response 8

Thank you for your feedback. We apologize for the use of vague terms in the experimental section. In the revised manuscript, we have provided the specific UV wavelength used, along with its intensity and any relevant details (365 nm, 0.245 W/cm²). We have also defined terms like C-ITXSP (The dormant species ITXSP was inoculated onto the surface of the fiber-based paper) and CA(PBS) (0.5 M, pH=8 phosphate buffer) to ensure clarity and understanding (Supporting Information P2 L50-55).

9. The pores shown in Fig. 1b-e are on a 10-100 μm scale, while those opened and closed by azobenzene are in the tens of nm. The authors should clarify how these different sizes of pores are formed. SEM or TEM images of the pores, especially before and after closure, are suggested.

Response 9

Thank you for your feedback. As shown in Figure 1(c,d), the grafting of the 3D mesh matrix fills the pores of the cellulose substrate itself. Since the BET detected the pores after being filled by GMA/4,4'-AZO, they are much smaller compared to the pure paper substrate. To further observe the distribution of GMA/4,4'-AZO/CA in EcoLeaf, we acquired 3D tomograms of EcoLeaf using LSCM. As can be clearly seen from Figure S2, GMA/4,4'-AZO, which emits light green fluorescence, fully fills the reticulation of the fiber base (Supporting Information P8 L194-196).

Figure S2. 3D tomograms of EcoLeaf and the 3D maps from different top-down view angles under LSCM (0° , 60° , 90°)

Additionally, the stomata of EcoLeaf were visualized by high-resolution TEM in the revised manuscript. As shown in Figure S4, with the increase in TEM magnification, the stomatal structure distributed on the 3D mesh mechanism can be clearly seen, and the stomatal size is only in the nanometer scale (0 - 2 nm). This result further validates the pore size parameters calculated from the pore size distribution curves of BET, thus demonstrating the successful construction of stomata in EcoLeaf. However, the average change in pore size due to contraction and expansion of EcoLeaf as measured by BET was less than 0.02 nm, making it difficult to observe a significant difference from TEM. Although it was not possible to visualize the contraction and expansion of

stomata, the results of the BET and water contact angel test, the different efficiency of CO₂ capture and water transport reflected the regular change of stomata under different light conditions (P9 L212). Thank you for bringing this to our attention

Figure S4. High-resolution transmission electron micrograph (TEM) of EcoLeaf (scale: 5-100 nm)

10. An explanation is needed for the observed decrease in free CA activity when the temperature exceeds 40 °C.

Response 10

Thank you for your feedback. In our preliminary study, we have thoroughly investigated the literature in the field of CA immobilization. Since CA is a natural enzyme protein, it possesses a specific protein structure. When the temperature changes, the protein structure and vibration frequency of CA will change to different degrees, which will lead to changes in enzyme protein activity. In general, the optimal temperature of free CA is 35 °C, and its activity strongly dependent on temperature. In addition, different immobilized substrates and immobilization methods have different effects on the CA structure, thus possessing different optimal temperatures^{22, 23, 24}. In the revised manuscript, we have included relevant discussion to explain this phenomenon (P8 L184-186). Thank you for bringing this to our attention.

Reviewer #2 (Remarks to the Author):

The work presented in the manuscript entitled "Leaves from painting to reality" is intended to show the engineering of an artificial leaf (named "EcoLeaf") for CO₂ sequestration from the atmosphere, mimicking the function of plant leaves. The concept of engineering synthetic materials able to convert atmospheric CO₂ into valuable chemicals is keynote. Differently from the majority of previous works based on metal catalysts, or semiconductors, the authors engineered an artificial leaf where the activity of the material depends on the enzyme carbonic anhydrase (CA). EcoLeaf is able to harvest light and convert it into thermal energy for CA activity (1), regulate CO₂ sequestration rate by opening-closing pores into a light-responsive polymeric matrix (2), transport the product of the enzymatic reaction outside the leaf (3), and finally degrade into the environment (4). The experimental work has been detailed precisely, especially the synthesis and characterization of the light-responsive 3D polymer matrix embedding the CA enzyme.

Response

Thank you for taking the time to review our manuscript. We appreciate your understanding of the engineering concept behind our work, which involves the development of an artificial leaf called "EcoLeaf" for CO₂ sequestration by mimicking the function of plant leaves.

Question 1

Small technical points could be improved such as in line 98 ("3000 rpm"), citing Figure 4(a) in the main text of the manuscript (to the best of my reading I haven't found it), as well as better describing the light-trapping varying dye depths in the section "experiment".

Response 1

Thank you for pointing out those specific technical points. We have corrected the unit of measurement for the rotational speed to ensure accuracy (Supporting Information P1 L24). After reviewing our manuscript, we realized that there was an error in the figure numbering. We have corrected this mistake and ensure that the appropriate figure is cited in the text (P11 L274).

In the "experiment" section, we acknowledge that the description of the light-trapping varying dye depths could be better articulated. We have revised this section (Supporting Information P1 L26-31) and added the variation of K/S with ink/ethanol solution (Figure S3(a)) to provide clearer and more detailed explanations of our experimental approach (Supporting Information P8 L198).

Thank you for your valuable feedback.

Figure S3. (a) Variation of K/S with ink/ethanol solution

Question 2

Nevertheless, mimicking Nature does not necessarily mean attempting to reproduce artificially what Nature has already built; on the contrary it means getting inspired by Nature for developing/engineering solutions for the needs we want to tackle. For this reason, I think that the shaping of EcoLeaf as a plant leaf (lines 69 and 102) justifies the title of the manuscript ("Leaves from painting to reality") but it is not of key importance, as any surface able to harvest light, regulate CO₂ sequestration and transport, as well biodegrade could serve the purpose, as well as be more easily integrated with other materials for building complex bio-inspired composite materials.

Response 2

Thank you for your valuable insight regarding the concept of mimicking nature in this manuscript. In fact, as you say, the shape is not important, but rather the fact that there are some useful properties that can be artificially achieved, inspired by some of the unique features of natural leaves. This paper does realize "light capture, stomatal contraction and expansion, CO₂ capture and conversion, soil degradation" functions based on the natural leaves, and these functions are indeed very helpful for carbon capture. The reason why we have cropped this paper into the shape of a leaf and given it such a title is to further enhance the interest and attractiveness of the article and draw the attention of readers from a wide range of fields under the Nature Communications journals to the problem, based on the fact that the key functions have been adequately discussed.

Based on your feedback, we will explore alternative design possibilities that are equally effective in achieving the desired functionalities while offering potential advantages in terms of ease of

integration with other materials in future research. Thank you for highlighting this perspective.

Question 3

Moreover, in the text of the manuscript (e.g. line 314: "...interrupting photosynthesis", or line 346: "...photosynthesis products.") the concept of "photosynthesis" is introduced. In Nature photosynthesis converts inorganic carbon (CO₂) into organic compounds ("synthesis") by using light ("photo") as a source of energy to fuel the Calvin Benson-Bassham (CBB) cycle. Furthermore, in the introduction, the authors mentioned the work performed by Erb and coworkers who developed a new-to-nature synthetic pathway (Schwander et al. Science, 2016) for fixating CO₂ in vitro, i.e. "synthesis", and successfully achieved to power the cycle by light, that is "photo", in chloroplast mimics (Miller et al. Science, 2020) so to perform "photo" + "synthesis" that is "photosynthesis" in vitro.

Despite the fact that the interconversion of CO₂ and HCO₃⁻ catalyzed by the CA enzyme is fundamental in Nature to maintain the biological flux that takes place via CO₂ fixation by Rubisco, the study described in this manuscript lacks any further enzymatic step to convert CO₂ into C_n organic compounds, that is it lacks "synthesis". Therefore, I think that "CO₂ sequestration" should be used instead of "photosynthesis".

On the other hand, EcoLeaf is a first step towards the engineering of enzymatic artificial leaves, so I hope that this study could open doors for the engineering of "truly photosynthetic" artificial leaves in the future.

Response 3

We appreciate the feedback on the concept of "photosynthesis" in our manuscript. We have carefully reconsidered the use of the term "photosynthesis" in the text and revised it to ensure clarity and accuracy. Regarding the suggestion to use "CO₂ sequestration" instead of "photosynthesis," we see the merit in this suggestion given the specific focus of our study on the enzymatic sequestration of CO₂. We have made the necessary revisions to the text to reflect this more accurately (P3 L79, P12 L280, P13 L311).

Furthermore, we completely agree with your perspective that EcoLeaf represents a significant step towards the development of enzymatic artificial leaves, and we share your hope for the future engineering of truly photosynthetic artificial leaves. We have incorporated this sentiment into the manuscript to highlight the potential impact of our study on paving the way for future

advancements in this area (P18 L419-436).

To verify the scalability of the EcoLeaf bionic system, in this study, the carbon capture core organ CA was replaced with formate dehydrogenase (FDH), which is capable of converting CO₂ to formic acid (HCOOH), and its reducing coenzyme nicotinamide adenine dinucleotide (NADH).

The substances involved in the reaction system (pure NADH, NAD⁺, H₂CO₃, FDH, and HCOOH dissolved in deionized water) and the solid carbon-generating solution of the EcoLeaf were characterized by high-performance liquid permeation chromatography. As shown in Figure S9, the solid-carbon generating solution of EcoLeaf peaked at 8.05 min (NAD⁺) and 15.9 min (HCOOH), respectively. This demonstrates that CO₂ can be successfully converted to HCOOH by replacing CA with FDH and NADH in the EcoLeaf bionic system. As you can see, the EcoLeaf can be fully designed and open doors for the engineering of “truly photosynthetic” artificial leaves in the future (Supporting Information P11 L247-259).

Figure S9. Chromatograms of pure NADH, NAD⁺, H₂CO₃, FDH, HCOOH dissolved in deionized water and the generating solution of the EcoLeaf containing FDH, NADH. Mobile phase H₂SO₄.

Reviewer #3 (Remarks to the Author):

The paper entitled “Leaves from Painting to Reality” by Zhu et al. reported an artificial leaf with adjustable stomata for carbon capture. The authors further highlighted that Ecoleaf is easy to degrade in the ecosystem, making it a highly attractive green technology. However, it is not clear how the mimicking of the opening/closing of the stomata of artificial leaf is related to the performance of CO₂ sequestration from the atmosphere. In natural leaves, the stomata are closed to prevent excessive loss of moisture and opened for photosynthesis when water and sunlight are abundant. If the proposed artificial leaf is for CO₂ sequestration, there seems to be no reason for the closure of artificial stomata when under relative lower intensity of light irradiation. Furthermore, no evidence is presented to show the scalability and long-term cyclic performance of the artificial leaves. For example, the subsequent transportation of the HCO₃⁻ generated in the Ecoleaf needs to be considered for practical applications.

Response

Thank you for taking the time to review our manuscript and sharing your valuable comments. In the revised manuscript, we have added additional experiments and provided a more detailed explanation of how the mimicking of the expansion and contraction of the stomata in the artificial leaf is related to its performance in CO₂ sequestration (In Response 1 below). First, for CA, carbon capture is more efficient when the pore size expands and more stable when the pore size contracts. In practice, the expanded pore state is preferable during the reaction, and shrinking the pore at the time of transportation or when the reaction is not required is more conducive to maintaining the stability of the enzyme molecule. More importantly, our present work aims to build a platform to lay the foundation for the subsequent multi-enzyme cascade reaction combining carbon capture and utilization. In multi-enzyme cascade reactions, matching the carbon capture efficiency to the carbon conversion efficiency is crucial. When the carbon capture efficiency is much higher than the conversion efficiency of the subsequent carbon capture products, the accumulated carbon capture products will cause the purity of the final conversion products to decrease on the one hand and affect enzyme activity due to substrate inhibition on the other. When the carbon capture efficiency is much lower than the conversion efficiency of the subsequent carbon capture products, the efficiency of the multi-enzyme cascade reaction decreases due to the lack of sufficient substrate for the carbon conversion reaction. According to the study in

this paper, the adjustment of stomatal size can regulate the rate of carbon capture in a wide range, which can lay a good foundation for matching the rate of the subsequent multi-enzyme cascade reaction.

Additionally, we have included relevant data and discussions on scalability and the long-term performance of Ecoleaf (In Response 2 below). Thank you for bringing these considerations to our attention.

The detailed comments are as follows:

(1) What's the main purpose of the adjustable stomata in the Ecoleaf for improving the efficiency of carbon capture? Is there a link between the adjustable stomata and the efficiency of carbon capture? The author declares the CA would be stable under the closing state, but how do we adjust that during the practical application (365 nm)?

Response 1

Thank you for your feedback. In the revised manuscript, we explored the difference in carbon capture efficiency of the EcoLeaf when its stomata were in the expansion and contraction state. As depicted in Figure 5(b), the CO₂ capture rate of EcoLeaf was markedly reduced when exposed to 365 nm UV illumination, in contrast to the 450 nm. This phenomenon may be due to two reasons: first, it is attributed to the contraction of the mesh pores within the 3D fabric matrix under UV light excitation (Figure 4(d-e)), which obstructs the contact pathway between gaseous CO₂ and the core carbon-fixing enzyme CA; In addition, the redistribution of hydrophilic and hydrophobic groups under different wavelengths of light also affects the CO₂ capture efficiency of EcoLeaf. Detailed analysis in the main text (P11 L275-286, P13 L314-315).

Figure 5. (b) Effect of stomatal expansion and contraction on the carbon capture performance of

EcoLeaf

Furthermore, our previous studies have revealed that, apart from stomatal contraction traits, the water transport characteristics also have a significant impact on the carbon sequestration performance of CA. Therefore, we investigated the influence of stomatal expansion and contraction traits on the water transport properties of EcoLeaf. As shown in Figure 5(d), When the roots of EcoLeaf were in contact with a water source and the stomata were contracted (365 nm), a decrease in water content was observed compared to when the stomata were expanded (450 nm), indicating that the expansion and contraction states of the stomata have an impact on the water transport properties of EcoLeaf. With expansive stomata, driven by the osmotic pressure of the liquid, water molecules were more likely to pass through the pores of the 3D mesh fabric matrix to reach the leaf surface, providing an adequate source of protons for CA carbon sequestration. However, when the stomata of EcoLeaf were contracted, a reduction in its own water content occurred. This weakening was primarily due to the stomatal contraction, which increased the density of the three-dimensional mesh fabric matrix of EcoLeaf and opposed the liquid osmotic pressure, hindering the transport of water molecules. This result also suggests that when the stomata are in a contracted state, water mass transfer has a greater effect on leaf water content than water evaporation. As a result, this process disrupted the catalytic conversion reaction of CA with CO₂ by reducing the number of protons while maintaining CA stability. This finding further supports the conclusion in the Figure 5(b).

Figure 5. (d) Changes in water content of EcoLeaf with and without water supply (365 nm and 450 nm)

In practice, when subjected to 365 nm UV illumination, the three-dimensional mesh matrix of

EcoLeaf becomes denser due to the contraction of stomata, which provides a good encapsulation of the protein structure of CA. When stimulated by unfavorable factors, CA is able to better maintain its spatial structure under the binding effect of the three-dimensional mesh, thus possessing better stability. This is verified in the results and discussion of Figure 4(f-g) (P9 L212, P11 L257-261).

Overall, for CA, carbon capture is more efficient when the pore size expands and more stable when the pore size contracts. In practice, the expanded pore state is preferable during the reaction, and shrinking the pore at the time of transportation or when the reaction is not required is more conducive to maintaining the stability of the enzyme molecule. More importantly, our present work aims to build a platform to lay the foundation for the subsequent multi-enzyme cascade reaction combining carbon capture and utilization. In multi-enzyme cascade reactions, matching the carbon capture efficiency to the carbon conversion efficiency is crucial. When the carbon capture efficiency is much higher than the conversion efficiency of the subsequent carbon capture products, the accumulated carbon capture products will cause the purity of the final conversion products to decrease on the one hand and affect enzyme activity due to substrate inhibition on the other. When the carbon capture efficiency is lower than the conversion efficiency of the subsequent carbon capture products, the efficiency of the multi-enzyme cascade reaction decreases due to the lack of sufficient substrate for the carbon conversion reaction. According to the study in this paper, the adjustment of stomatal size can regulate the rate of carbon capture in a wide range, which can lay a good foundation for matching the rate of the subsequent multi-enzyme cascade reaction.

(2) Cyclic and long-term testing of the capability to CO₂ capture are not mentioned.

Response 2

Thank you for your insightful comment. In the revised manuscript, we have incorporate data and discussion on cyclic and long-term testing to evaluate the CO₂ capture capability of the Ecoleaf. As can be seen from Figure 5(e), the CO₂ capture ability (carbon sequestration rate) of EcoLeaf with water supply gradually stabilized over the long-term and cyclic (from cycle 1 to cycle 3) test without significant product inhibition. This is mainly attributed to its substance-transport properties. In addition, the CO₂ capture ability of EcoLeaf without water supply was unstable and gradually weakened due to the inhibition of products and proton sources. This further

demonstrates that the material transport property enables the EcoLeaf prepared in this study to maintain a stable carbon sequestration capacity over a long period of time (P13 L318-320; P15 L353-362).

Figure 5. (e) Long-term carbon sequestration stability of EcoLeaf with and without water supply and cycling stability of EcoLeaf with water supply

(3) For the LCSM, please give the results for the fluorescence intensity of FITC with the depth. It is very important to prove the distribution of the CA in the Eco-leaf.

Response 3

Thank you for your feedback. In the revised manuscript, we have included detailed results and discussion on the fluorescence intensity of three-dimensional mesh substrate and FITC-CA at different depths within the Ecoleaf. Figure S2 displays the 3D tomograms of EcoLeaf and the 3D maps from different top-down view angles (0°, 60°, 90°) (Supporting Information P8 L194-196). The encapsulation of the fiber surface by the 3D mesh matrix and the filling of the fiber pores can be clearly seen in the figure, reinforcing the successful construction of the 3D mesh cloth matrix. Additionally, the bright green fluorescence (FITC-CA) exhibits a relatively uniform distribution within the 3D mesh matrix.

Figure S2. 3D tomograms of EcoLeaf and the 3D maps from different top-down view angles under LCSM (0°, 60°, 90°)

(4) Figure 3f has three diagrams. They should be labelled in 3(f), 3(g) and 3(h). Why there are two bars (results) for 365nm in 3f? It appears that the differences the temperature stability and pH stability under 365 and 450 nm irradiation are not so large. Are they really significant?

Response 4

Thank you for pointing out the missing labels in Figure 3. We have added the labeling as per your suggestion (P9 L212).

Regarding your question, the two bars in under 365 nm irradiation represent a cycle of excitation (365 nm, 450 nm, 365 nm). The purpose of this data is to explore whether the expansion and contraction of the stomatal are reversible through a loop. In the revised manuscript, we added the effect of stomatal expansion and contraction characteristics on CO₂ capture performance. As depicted in Figure 5(b), the CO₂ capture rate of EcoLeaf was markedly reduced when exposed to 365 nm UV illumination (stomatal contraction), in contrast to the 450 nm (stomatal expansion). This enables the reader to more intuitively access the core findings of this study.

Although the magnitude of the effect of hole changes on stability is small, it does have some effect. This effect can provide some assistance in the preservation of CA, as well as further feedback on the process of hole size change. More importantly, the contraction and expansion of stomata are essential for the regulation of carbon sequestration rates. This law provided an important reference for the design of the subsequent multi-enzyme cascade reaction. In the process of carbon

sequestration in natural leaves, CO₂ needs to be converted into value-added products through the dark reaction stage after capture. For EcoLeaf, our present work aims to build a platform to lay the foundation for the subsequent multi-enzyme cascade reaction. In multi-enzyme cascade reactions, matching the carbon capture efficiency to the carbon conversion efficiency is crucial. When the carbon capture efficiency is much higher than the conversion efficiency of the subsequent carbon capture products, the accumulated carbon capture products will cause the purity of the final conversion products to decrease on the one hand and affect enzyme activity due to substrate inhibition on the other. When the carbon capture efficiency is much lower than the conversion efficiency of the subsequent carbon capture products, the efficiency of the multi-enzyme cascade reaction decreases due to the lack of sufficient substrate for the carbon conversion reaction. According to the study in this paper, the adjustment of stomatal size can regulate the rate of carbon capture in a wide range, which can lay a good foundation for matching the rate of the subsequent multi-enzyme cascade reaction.

(5) The sunlight contains both 365 and 450 nm light, so how sunlight regulates the opening and closing of stomata on the Ecoleaf needs to be clearly explained.

Response 5

Thank you for your feedback. In the revised manuscript, we have conducted additional experiments to explore this. As shown in Figure S7, when EcoLeaf was irradiated under UV light (365 nm, 20.3 mW/cm²) for 20 min and then transferred to darkness, blue light (450 nm, 19.1 mW/cm²), and natural sunlight (27.5–28.8 mW/cm²), respectively, the absorbance ($\lambda=390$ nm) decreased, and the decline rate of absorbance increased in order (Supporting Information P10 L222-240). Among them, the slow expansion of stomata in the dark environment was due to the spontaneous transition of GMA/4,4'-AZO from the *cis* form of the substable state to the *trans* form of the steady state (Figure 4(a)). It can be seen that blue light is able to accelerate the process, and thus the blue light group possesses a higher rate of decrease. The sudden increase in this group was mainly due to the fluctuation of natural light, but the overall decreasing trend was still able to remain stable. In addition, the expansion and contraction of GMA/4,4'-AZO stomata are directly related to light intensity. The greater the intensity of blue light, the greater the stomatal expansion. Of the solar radiation that reaches the earth's surface from the sun, visible light accounts for about 40% of the total solar radiation, infrared light for about 50%, and ultraviolet light for only about

10%. Since visible light is more intense, the mesh of EcoLeaf tends to expand under natural sunlight. A comparison of the red and blue lines in Figure S7 shows that the natural light group has a more significant effect on stomatal expansion because it possesses a stronger light intensity than the blue light group. This demonstrates the dependence of stomatal expansion and contraction on light intensity. We appreciate your valuable feedback and suggestions.

Figure S7. Changes in absorbance (pore properties) of EcoLeaf under darkness, 450 nm blue light (19.1 mW/cm^2), and natural sunlight ($27.5\text{--}28.8 \text{ mW/cm}^2$) after irradiation for 20 min at 365 nm UV light (20.3 mW/cm^2)

(6) More measurements of the capture performance of the Ecoleaf need to be provided. The results in Figure 4d just show the short-time absorption of the CO_2 .

Response 6

Thank you for your feedback. In the revised manuscript, we have incorporate data and discussion on cyclic and long-term testing to evaluate the CO_2 capture capability of the Ecoleaf. As can be seen from Figure 5(e), the CO_2 capture ability (CO_2 capture rate) of EcoLeaf with water supply gradually stabilized over the long-term and cyclic test (from cycle 1 to cycle 3) without significant product inhibition. This is mainly attributed to its substance-transport properties. In addition, the CO_2 capture ability of EcoLeaf without water supply was unstable and gradually weakened due to the inhibition of products and proton sources. This further demonstrates that the material transport property enables the EcoLeaf prepared in this study to maintain a stable carbon sequestration capacity over a long period of time. Detailed analysis is in the main text (P15 L353-362).

Figure 5. (e) Long-term carbon sequestration stability of EcoLeaf with and without water supply and cycling stability of EcoLeaf with water supply.

(7) What's the difference in Ecoleaf absorption of CO₂ performance under different light irradiation?

Response 7

Thank you for your insightful feedback. We have added two crucial characterizations. First, as depicted in Figure 5(b), the CO₂ capture rate of EcoLeaf was markedly reduced when exposed to 365 nm UV illumination, in contrast to the 450 nm. This phenomenon may be due to two reasons: first, it is attributed to the contraction of the mesh pores within the 3D fabric matrix under UV light excitation (Figure 4(d-e)), which obstructs the contact pathway between gaseous CO₂ and the core carbon-fixing enzyme CA; In addition, the redistribution of hydrophilic and hydrophobic groups under different wavelengths of light also affects the CO₂ capture efficiency of EcoLeaf. Detailed analysis in the main text (P12 L273-286).

Figure 5. (b) Effect of stomatal expansion and contraction states on the carbon capture performance of EcoLeaf

Furthermore, our previous studies have revealed that, apart from stomatal contraction traits, the water transport characteristics also have a significant impact on the carbon sequestration performance of CA. Therefore, we investigated the influence of stomatal expansion and contraction traits on the water transport properties of EcoLeaf. As shown in Figure 5(d), When the roots of EcoLeaf were in contact with a water source and the stomata were contracted (365 nm), a decrease in water content was observed compared to when the stomata were expanded (450 nm), indicating that the expansion and contraction states of the stomata have an impact on the water transport properties of EcoLeaf. With expansive stomata, driven by the osmotic pressure of the liquid, water molecules were more likely to pass through the pores of the 3D mesh fabric matrix to reach the leaf surface, providing an adequate source of protons for CA carbon sequestration. However, when the stomata of EcoLeaf were contracted, a reduction in its own water content occurred. This weakening was primarily due to the stomatal contraction, which increased the density of the three-dimensional mesh fabric matrix of EcoLeaf and opposed the liquid osmotic pressure, hindering the transport of water molecules. This result also suggests that when the stomata are in a contracted state, water mass transfer has a greater effect on leaf water content than water evaporation. As a result, this process disrupted the catalytic conversion reaction of CA with CO₂ by reducing the number of protons while maintaining CA stability. This finding further supports the conclusion in the Figure 5(b).

Figure 5. (d) Changes in water content of EcoLeaf with and without water supply (365 nm and 450 nm)

(8) Considering the low solubility and stability of H₂CO₃, is it effective to migrate the H₂CO₃ from the Ecoleaf to the bulk water for long-term operation? Repeated testing is required.

Response 8

Thank you for your feedback. We have conducted repeated testing to evaluate the effectiveness of migrating H_2CO_3 from the Ecoleaf to bulk water for long-term operation, considering its low solubility and stability. As can be seen from Figure 5(e), the ability (CO_2 capture rate) of EcoLeaf with water supply gradually stabilized over the long-term and cycling test (from cycle 1 to cycle 3) without significant product inhibition (P13 L319-320, P15 L353-362). This is mainly attributed to its substance-transport properties. In addition, the CO_2 capture ability of EcoLeaf without water supply was unstable and gradually weakened due to the inhibition of products H_2CO_3 and proton sources. This further demonstrates that the material transport property enables the EcoLeaf prepared in this study to maintain a stable carbon sequestration capacity over a long period of time.

Figure 5. (e) Long-term carbon sequestration stability of EcoLeaf with and without water supply and cycling stability of EcoLeaf with water supply

(9) Was the enzyme transferred to the bulk solution with the H_2CO_3 during the regeneration process?

Response 9

Thank you for your comment. This issue is crucial to our research. Therefore, we double-checked whether the enzyme would leak by conducting two further experiments.

Experiment 1: Roots of EcoLeaf containing FITC-CA were placed in an aqueous solution, and samples were taken at 5-minute intervals. The aqueous solution was observed under LSCM for the appearance of fluorescence.

Experiment 2: Roots of EcoLeaf containing CA were placed in an aqueous solution, and samples were taken every 10 minutes, followed by staining of the samples with Caumas Brilliant Blue

Rapid Staining Solution G-250. The absorbance of the samples was measured under a UV-vis spectrophotometer.

The results of the experiment are shown below, and the strong fluorescence of FITC was not observed under LSCM. In addition, CA leakage rates remain extremely low (Figure S8). It can thus be demonstrated that no enzyme leakage occurs when CA is encapsulated through the 3D mesh cloth matrix. The detailed analysis is shown in the text (P15 L363-371; Supporting Information P11 L241-245).

Figure S8. (a) Aqueous solutions attached to petioles of EcoLeaf containing FITC-CA were observed under LSCM (time interval: 5 min); (b) The CA leakage rate in the aqueous solution (attached to the petioles of EcoLeaf) stained by G-250 was measured by a UV-vis spectrophotometer (time interval: 10 min)

(10) Please provide more characterizations of the Eco-leaf, such as its surface pores structure under high-resolution SEM, mechanical strength, etc.

Response 10

Thank you for your advice. The stomata of EcoLeaf were visualized by high-resolution TEM in the revised manuscript. As shown in Figure S4 (SI P9 L206-207), with the increase in TEM magnification, the stomatal structure distributed on the 3D mesh mechanism can be clearly seen, and the stomatal size is only in the nanometer scale (0–2 nm). This result further validates the pore size parameters calculated from the pore size distribution curves of BET. However, the average change in pore size due to contraction and expansion of EcoLeaf as measured by BET was less than 0.02 nm, making it difficult to observe a significant difference under TEM. Although it was not possible to visualize the contraction and expansion of stomata, the results of the BET test, the pattern of change in the optical contact angle, CO₂ capture efficiency, CA stability, and water

cycle characterization reflected the regular change of stomata under different light conditions..

Figure S4. High-resolution transmission electron micrograph (TEM) of EcoLeaf (scale: 5-100 nm)

In addition, we conducted stress-strain tests on five natural blades from different sources and two EcoLeaves using a multifunctional testing machine to provide data to support our mechanical performance requirements. The test results are shown in the Figure 2, comprising lignocellulose, the mechanical properties of the paper used to make EcoLeaf are higher than those of most natural leaves but similar to palm fronds. The detailed experimental procedure and result analysis are shown in the main text (P7 L158-177). Thank you for bringing this to our attention.

Figure 2(a) Natural leaves from *Trachycarpus fortunei*, *Cercis chinensis bunge*, *Pyrus ussuriensis*, *Zamioculcas zamiifolia engl*, *Holly*, and EcoLeaf respectively; (b) Stress-strain curves and (c) Young's modulus of natural leaves from different sources and EcoLeaf

- Ojha N, Kumar S. Tri-phase photocatalysis for CO₂ reduction and N₂ fixation with efficient electron transfer on a hydrophilic surface of transition-metal-doped MIL-88A (Fe). *Appl Catal B-Environ* **292**, 120166 (2021).
- Kamal Hussien M, *et al.* Metal-free four-in-one modification of g-C₃N₄ for superior photocatalytic CO₂ reduction and H₂ evolution. *Chem Eng J* **430**, 132853 (2022).
- Bhattacharjee S, *et al.*

- Photoelectrochemical CO₂-to-fuel conversion with simultaneous plastic reforming. *Nat Synth* **2**, 182-192 (2023).
4. Wang Y, Zhou Q, Zhu Y, Xu D. High efficiency reduction of CO₂ to CO and CH₄ via photothermal synergistic catalysis of lead-free perovskite Cs₃Sb₂I₉. *Appl Catal B-Environ* **294**, (2021).
 5. Wang X, *et al.* Immobilizing perovskite CsPbBr₃ nanocrystals on Black phosphorus nanosheets for boosting charge separation and photocatalytic CO₂ reduction. *Appl Catal B-Environ* **277**, 119230 (2020).
 6. Mu YF, Zhang W, Dong GX, Su K, Zhang M, Lu TB. Ultrathin and Small-Size Graphene Oxide as an Electron Mediator for Perovskite-Based Z-Scheme System to Significantly Enhance Photocatalytic CO₂ Reduction. *Small* **16**, (2020).
 7. Bera S, Shyamal S, Pradhan N. Chemically Spiraling CsPbBr₃ Perovskite Nanorods. *J Am Chem Soc* **143**, 14895-14906 (2021).
 8. Wang Q, *et al.* Coupling CsPbBr₃ Quantum Dots with Covalent Triazine Frameworks for Visible-Light-Driven CO₂ Reduction. *ChemSusChem* **14**, 1131-1139 (2021).
 9. Wang X, Wang Z, Li Y, Wang J, Zhang G. Efficient photocatalytic CO₂ conversion over 2D/2D Ni-doped CsPbBr₃/Bi₃O₄Br Z-scheme heterojunction: Critical role of Ni doping, boosted charge separation and mechanism study. *Appl Catal B-Environ* **319**, (2022).
 10. Yuan S-X, Su K, Feng Y-X, Zhang M, Lu T-B. Lattice-matched in-situ construction of 2D/2D T-SrTiO₃/CsPbBr₃ heterostructure for efficient photocatalysis of CO₂ reduction. *Chinese Chem Lett* **34**, (2023).
 11. Que M, *et al.* Anchoring of Formamidinium Lead Bromide Quantum Dots on Ti₃C₂ Nanosheets for Efficient Photocatalytic Reduction of CO₂. *ACS Appl Mater & Interfaces* **13**, 6180-6187 (2021).
 12. Dong G-X, Zhang W, Mu Y-F, Su K, Zhang M, Lu T-B. A halide perovskite as a catalyst to simultaneously achieve efficient photocatalytic CO₂ reduction and methanol oxidation. *Chem Commun* **56**, 4664-4667 (2020).
 13. Wu S, *et al.* High light-to-fuel efficiency and CO₂ reduction rates achieved on a unique nanocomposite of Co/Co doped Al₂O₃ nanosheets with UV-vis-IR irradiation. *Energ & Environ Sci* **12**, 2581-2590 (2019).
 14. Sheng J, *et al.* Identification of Halogen-Associated Active Sites on Bismuth-Based Perovskite

- Quantum Dots for Efficient and Selective CO₂-to-CO Photoreduction. *ACS Nano* **14**, 13103-13114 (2020).
15. Hu C, *et al.* Near-infrared-featured broadband CO₂ reduction with water to hydrocarbons by surface plasmon. *Nat Commun* **14**, (2023).
 16. Zheng Q, *et al.* Surface Halogen Compensation on CsPbBr₃ Nanocrystals with SOBr₂ for Photocatalytic CO₂ Reduction. *ACS Mater Lett* **4**, 1638-1645 (2022).
 17. Jia J, *et al.* Visible and Near-Infrared Photothermal Catalyzed Hydrogenation of Gaseous CO₂ over Nanostructured Pd@Nb₂O₅. *Adv Sci* **3**, (2016).
 18. Chen Z, *et al.* Boosting Photocatalytic CO₂ Reduction on CsPbBr₃ Perovskite Nanocrystals by Immobilizing Metal Complexes. *Chem Mater* **32**, 1517-1525 (2020).
 19. Zhang G, *et al.* Interfacial Engineering of Semicohherent Interface at Purified CsPbBr₃ Quantum Dots/2D-PbSe for Optimal CO₂ Photoreduction Performance. *ACS Appl Mater Interfaces* **14**, 44909-44921 (2022).
 20. Tahir M, Tahir B. Constructing S-scheme 2D/0D g-C₃N₄/TiO₂ NPs/MPs heterojunction with 2D-Ti₃AlC₂ MAX cocatalyst for photocatalytic CO₂ reduction to CO/CH₄ in fixed-bed and monolith photoreactors. *J Mater Sci Technol* **106**, 195-210 (2022).
 21. Fu G, *et al.* Rh/Al Nanoantenna Photothermal Catalyst for Wide-Spectrum Solar-Driven CO₂ Methanation with Nearly 100% Selectivity. *Nano Lett* **21**, 8824-8830 (2021).
 22. Shen J, Yuan Y, Salmon S. Carbonic Anhydrase Immobilized on Textile Structured Packing Using Chitosan Entrapment for CO₂ Capture. *ACS Sustainable Chem Eng* **10**, 7772-7785 (2022).
 23. Chang S, He Y, Li Y, Cui X. Study on the immobilization of carbonic anhydrases on geopolymer microspheres for CO₂ capture. *J Clean Prod* **316**, (2021).
 24. Zaidi S, Srivastava N, Kumar Khare S. Microbial carbonic anhydrase mediated carbon capture, sequestration & utilization: A sustainable approach to delivering bio-renewables. *Bioresource Technol* **365**, (2022).

Reviewers' Comments:

Reviewer #1:

Remarks to the Author:

[Note from the Editor: Reviewer #1 was asked to also review the response given to the original Reviewer #3.]

Comments for authors (Reviewer 1)

The manuscript has significantly improved, with the authors incorporating additional experiments to bolster their innovative approach. The inclusion of a table summarizing state-of-the-art CO₂ conversion systems, highlighting key factors such as material cost, preparation method, light utilization efficiency, and potential environmental impact, offers valuable insights into both the current study and future innovations in the field.

The authors have adequately addressed all of my comments, and based on the comprehensive nature of their work, I recommend the manuscript for publication in Nature Communications. However, I have one suggestion:

While the authors have conducted detailed experiments comparing the mechanical properties of EcoLeaf with those of natural leaves, it would be advantageous to underscore the significance of the material's mechanical robustness in either the abstract, introduction, or conclusion section. By emphasizing this aspect alongside its biodegradability, the authors can further highlight the positive environmental impact of their work.

Comments for authors (Reviewer 3)

The authors present compelling evidence illustrating the long-term CO₂ absorption behavior, the role of sunlight in facilitating stomatal opening, and the stabilization of CO₂ absorption over multiple cycles, effectively addressing most of the concerns raised in review 3.

In Figure 5d, the authors demonstrate a reduction in water content when the EcoLeaf stomata are contacted. Additionally, they show that the EcoLeaf's water content decreases when the stomata expands without water supply. However, despite observing this phenomenon, it appears that the EcoLeaf system does not leverage the reduced water content during the contracted state.

My question is whether the change in water content of EcoLeaf in the contacted state (365nm) is less significant than that of the Leaf in the expanded state when there is no water supply.

Furthermore, I question whether the contraction of pores, which prevents water loss in the system, aids in encapsulating the protein structure of CA and thereby enhances stability. This data would be valuable to include in Figure 5d.

Minor comment: could you clarify the meaning of "(3 parallel groups)" as mentioned in P13 line 315?

Reviewer #2:

Remarks to the Author:

The authors have addressed my minor points, as well as I particularly appreciated that they reconsidered the concept of photosynthesis (major point), and they substituted the term "photosynthesis" with "CO₂ sequestration" throughout the text of the manuscript.

I have still found some typos in the text, see lines 333 and 339 "contracted".

In the revised version of the paper the authors have now mentioned the use of the enzyme formate dehydrogenase for "scalability". This is unclear to me. The manuscript can be accepted for publication after clarifying this point.

Reviewer #1 (Remarks to the Author):

Comments for authors (Reviewer 1)

The manuscript has significantly improved, with the authors incorporating additional experiments to bolster their innovative approach. The inclusion of a table summarizing state-of-the-art CO₂ conversion systems, highlighting key factors such as material cost, preparation method, light utilization efficiency, and potential environmental impact, offers valuable insights into both the current study and future innovations in the field. The authors have adequately addressed all of my comments, and based on the comprehensive nature of their work, I recommend the manuscript for publication in *Nature Communications*.

Response

Thank you for once again taking the time to review our revised manuscript. We are glad to hear that the manuscript has significantly improved. We appreciate your suggestions and have carefully revised the manuscript.

Question 1

While the authors have conducted detailed experiments comparing the mechanical properties of EcoLeaf with those of natural leaves, it would be advantageous to underscore the significance of the material's mechanical robustness in either the abstract, introduction, or conclusion section. By emphasizing this aspect alongside its biodegradability, the authors can further highlight the positive environmental impact of their work.

Response 1

Thank you for your valuable comments. In the revised version, we have added the importance of the material's mechanical properties alongside its biodegradability in the Abstract (P1 L17-20), Introduction (P4 L97-99), and Conclusion sections (P18 L428-430).

Comments for authors (Reviewer 1)

The authors present compelling evidence illustrating the long-term CO₂ absorption behavior, the role of sunlight in facilitating stomatal opening, and the stabilization of CO₂ absorption over multiple cycles, effectively addressing most of the concerns raised in review 3.

Response

Thank you for your feedback. We are grateful for your positive evaluation.

Question 1

In Figure 5d, the authors demonstrate a reduction in water content when the EcoLeaf stomata are contracted. Additionally, they show that the EcoLeaf's water content decreases when the stomata expand without water supply. However, despite observing this phenomenon, it appears that the EcoLeaf system does not leverage the reduced water content during the contracted state. My question is whether the change in water content of EcoLeaf in the contracted state (365nm) is less significant than that of the Leaf in the expanded state when there is no water supply. Furthermore, I question whether the contraction of pores, which prevents water loss in the system, aids in encapsulating the protein structure of CA and thereby enhances stability. This data would be valuable to include in Figure 5d.

Response 1

Thank you for your feedback. We have carefully considered the issues you raised. In the revised manuscript, we have included the change in water content of EcoLeaf (365 nm) in the absence of water supply in Figure 5d (P13 L314). As shown in Figure 5(d), when there was no water supply, the rate of water loss (k_2) of EcoLeaf in the pore-contracting group (365 nm) was slightly higher than that of the pore-expanding group (450 nm, k_1) in the primary stage. This may be due to the contraction process of the stomata induced by the UV light at 365 nm, which squeezed some of the water outside the EcoLeaf. Importantly, the decreasing rate of k_2 value was much lower than that of k_1 . This is mainly attributed to the water retention effect caused by the contraction of the mesh, which prevents the volatilization of water molecules retained inside the mesh. As a result, the water content of the contracted group has become higher than the expanded group in the later stage. This phenomenon may also help to retain the protein structure of CA and thereby enhance stability as you indicated.

Figure 4 (d) Changes in water content of EcoLeaf with and without water supply (365 nm and 450 nm)

Question 2

Minor comment: could you clarify the meaning of "(3 parallel groups)" as mentioned in P13 line 315?

Response 2

The label "(3 parallel groups)" in P13 Line317 represents that we prepared three sets of parallel samples and examined their carbon sequestration properties in order to echo the red and gray error areas in Figure 5 (b). In order to enhance the rigor of the figure notes, we have modified all the relevant labels in the text. (P7 L163, P13 L316, P13 L321)

Reviewer #2 (Remarks to the Author):

The authors have addressed my minor points, as well as I particularly appreciated that they reconsidered the concept of photosynthesis (major point), and they substituted the term “photosynthesis” with “CO₂ sequestration” throughout the text of the manuscript.

Response

Thank you for taking the time to review our manuscript again. We are pleased that our revisions were able to address your concerns.

Question 1

I have still found some typos in the text, see lines 333 and 339 “contracted”.

Response 1

Thank you for bringing this to our attention. We apologize for any oversight in our proofreading. We have carefully reviewed the manuscript and corrected the typos.

Question 2

In the revised version of the paper the authors have now mentioned the use of the enzyme formate dehydrogenase for “scalability”. This is unclear to me. The manuscript can be accepted for publication after clarifying this point.

Response 2

Thank you for your feedback. In fact, formate dehydrogenase was introduced in this study to further elucidate the designability of the EcoLeaf system. We apologize for using inappropriate words in the manuscript. In order to enhance the rigor of the language, we have modified "scalability" to "designability," implying that CA can be replaced by other enzymes (alcohol dehydrogenase, formaldehyde dehydrogenase, etc.) to achieve richer product conversion (P15 L380, L382). It is hoped that this study will provide a platform for biological carbon sequestration and conversion.

Reviewers' Comments:

Reviewer #1:

Remarks to the Author:

All the comments from reviewers 1 and 3 have been addressed. However, there is a minor issue with the labeling in Figure 5d, as it's not correctly ordered. Once this labeling is fixed, the manuscript is recommended for publication.

Reviewer #2:

Remarks to the Author:

The manuscript has improved significantly.

The authors have addressed my comments, and I support publication in Nature Communications.

REVIEWERS' COMMENTS

Reviewer #1 (Remarks to the Author):

All the comments from reviewers 1 and 3 have been addressed. However, there is a minor issue with the labeling in Figure 5d, as it's not correctly ordered. Once this labeling is fixed, the manuscript is recommended for publication.

Response

Thank you for once again taking the time to review our manuscript. We have adjusted the order of the figure notes in Figure 5(d).

Figure 5(d) Changes in water content of EcoLeaf with and without water supply (365 nm and 450 nm)

Reviewer #2 (Remarks to the Author):

The manuscript has improved significantly.

The authors have addressed my comments, and I support publication in Nature Communications.

Response

Thank you for once again taking the time to review our manuscript. We are very happy to see your approval.